


# A new look at the multi-G model for organic carbon degradation in surface marine sediments for coupled benthic-pelagic simulations of the global ocean

Konstantin Stolpovsky[1], Andrew W. Dale[1], Klaus Wallmann[1]

[1] GEOMAR Helmholtz Centre for Ocean Research Kiel, Kiel, Germany

*Correspondence to*: Konstantin Stolpovsky (kstolpovsky@geomar.de)

**Abstract.** The kinetics of particulate organic carbon (POC) mineralization in marine surface sediments is not well constrained. This creates considerable uncertainties when benthic processes are considered in global biogeochemical or Earth system circulation models to simulate climate-ocean interactions and biogeochemical tracers in the ocean. In an attempt to improve our understanding of the rate and depth distribution of organic carbon mineralization in bioturbated (0 – 10 cm) sediments, we parameterized a 1-D diagenetic model that simulates the reactivity of three discrete POC pools at global scale (a 'multi-G' model). The rate constants of the three reactive classes (highly reactive, reactive, refractory) are fixed and determined to be 70 yr$^{-1}$, 0.5 yr$^{-1}$, and ~0.001 yr$^{-1}$, respectively, based on the Martin curve model for pelagic POC degradation. In contrast to previous approaches, the reactivity of the organic material degraded in the seafloor is continuous with, and set by, the apparent reactivity of material sinking through the water column. The model is able to simulate a global database (185 stations) of benthic oxygen and nitrate fluxes across the sediment-water interface in addition to porewater oxygen and nitrate distributions and organic carbon burial efficiencies. It is further consistent with degradation experiments of fresh phytoplankton. We propose that an important yet mostly overlooked consideration in previous upscaling approaches is the *proportion* of the relative reactive POC classes reaching the seafloor in addition to their reactivity. The approach presented is applicable to both steady-state and non-steady state scenarios, and links POC degradation kinetics in sedimentary environments to water depth and the POC rain rate to the seafloor.

## 1 Introduction

Mineralization and burial of particulate organic carbon (POC) in marine sediments is a core component of the Earth's carbon cycle, helping to regulate atmospheric oxygen and carbon dioxide levels over glacial time scales and longer (Berner and Canfield, 1989; Mackenzie et al., 2004). On time scales relevant to society and nutrient oceanic residence times ($10^3$ yr), the recycling of POC in the seafloor leads to a loss of fixed N and P from the ocean, thus modulating basin-wide nutrient inventories (Van Cappellen, 2003). Global biogeochemical box or circulation models (collectively termed here Earth System Models, ESM) that are designed to study e.g. climate driven environmental changes or proxies in sediment archives, thus rely on knowledge of the fraction of POC reaching the seafloor that is mineralized, and the fraction that is buried.



Earth System Models deal with the recycling of biogenic material in sediment in a variety of ways, from the very simple (ignoring it completely) to the more complex yet computationally expensive coupling with 1-D vertically resolved or layered diagenetic models (e.g. Munhoven, 2007). Intermediate approaches make use of computationally efficient empirical transfer functions (vertically integrated models) to simulate benthic feedbacks of the N, P, Fe and $O_2$ cycles (Middelburg et al., 1997;

Soetaert et al., 2000; Capet et al., 2016; Bohlen et al., 2012; Somes et al., 2013; Dale et al., 2015; Wallmann, 2010; Tschumi et al., 2011; Yang and Gruber, 2016). Feedbacks of nutrients computed using these functions may treat POC degradation implicitly and be independent of one another. Whilst transfer functions undoubtedly offer a powerful and pragmatic approach, 1-D diagenetic models provide a more complete coupling between carbon and nutrient cycles and their fluxes across the seafloor (Soetaert et al., 2000).

In 1-D models, the depth in the sediment over which POC degrades is usually determined by the reactivity or 'freshness' of the material and its net downward movement due to burial and bioturbation (mixing by animals). It is important to gain an improved understanding of these processes because the depth over which POC is mineralized in sediments strongly determines the flux of redox-sensitive elements (e.g. oxygen, nitrate, phosphate, iron etc.) across the sediment surface (Stolpovsky et al., 2015). Yet, it is notoriously difficult to prescribe the reactivity of POC once it arrives at the seafloor due

the complex interaction of many ecosystem properties (Bianchi et al., 2016). These include mineral interactions, geopolymerization, thermodynamic factors and the activity of micro-and macro-biota (e.g. de Leeuw and Largeau, 1993; Aller, 1994; Mayer, 1995; LaRowe and Van Cappellen, 2011). The net result is an apparent reactivity of POC that decreases with time, often parameterized in diagenetic models using the reactive continuum model (Middelburg, 1989; Boudreau and Ruddick, 1991). Implementation of POC dynamics for continuum models in bioturbated sediments is possible, yet somewhat

cumbersome (Dale et al., 2015). Recently, Stolpovsky et al. (2015) successfully parameterized a simple continuum model for bioturbated sediments as a function of the POC rain rate to the seafloor using a global database of in situ flux measurements. Several current generation ESMs rely on constitutive equations that describe linear degradation kinetics of one or more sedimentary POC fractions, or 'G', as a function of e.g. burial rate (reviewed by Arndt et al., 2013). Whilst this so-called multi-G model (Jørgensen, 1978) works well for individual sites where POC degradation can be constrained against

porewater and solid phase concentrations, there is no evidence that the transferability of these models is guaranteed at the global scale. In fact, building upon previous work (Toth and Lerman, 1977; Emerson et al., 1985; Tromp et al., 1995; Boudreau, 1997; Boudreau and Ruddick, 1991), Arndt et al. (2013) showed that no statistically relationship exists between the rate constants for POC degradation and major controlling factors such as burial rate and water depth. This severely limits the predictive capacity of EMSs to faithfully reproduce, and draw conclusions about, benthic feedbacks and the role of

sediments in global biogeochemical cycles of carbon, nutrients and other non-conservative tracers.

In this study, we present a novel methodology to constrain the rate constants of discrete organic matter reactive classes in sediments using the classic multi-G approach. Our motivation is to provide a more realistic POC degradation model that can be coupled to ESMs. We employ the same global database of benthic $O_2$ and $NO_3^-$ fluxes to constrain the model that we used previously to derive the continuum model (Stolpovsky et al., 2015). The model is further consistent with global POC burial





rates and porewater $O_2$ and $NO_3^-$ distributions. In contrast to all previous approaches (to the best of our knowledge), the apparent reactivity of the organic material degraded in the seafloor is continuous with, and set by, the apparent reactivity of material sinking through the water column. This is achieved by assuming that the efficiency of carbon transfer to the ocean interior and to the seafloor (i.e. the biological pump; Sarmiento and Gruber, 2006) can be approximated using a power law,

that is, the Martin curve (Martin et al., 1987). We find that the relative proportion of the reactive classes reaching the seafloor allows for transferability, upscaling and extrapolation of the model to the global scale. The coherence between POC degradation in the water column and sediments should provide a more realist estimate of benthic pelagic coupling of redox sensitive elements (e.g. $O_2$, $NO_3^-$, $PO_4^{3-}$, $Fe^{2+}$) in ESMs that explicitly account for seafloor mineralization processes.

## 2 Model Structure

In this section, we first describe how the apparent reactivity ($k_{app}$ in $yr^{-1}$) of POC sinking through the water column is calculated. This information is then used to define the rate constants and fractions of three reactive POC pools in bioturbated surface sediments (upper 10 cm) by simulating global databases of $O_2$ and $NO_3^-$ fluxes using a reaction-transport sediment model. This latter model is based on our previous study where POC degradation rates were defined using a reactive continuum approach (Stolpovsky et al., 2015).

### 2.1 Apparent reactivity of organic matter sinking through the water column

The apparent reactivity of POC raining to the seafloor was calculated by considering the sinking flux of POC. This was achieved by simulating the Martin curve (Martin et al., 1987) that has been widely used to describe the flux, or rain rate, of sinking POC in water column as a function of water depth, $wd$ (RRPOC):

$$\frac{\text{RRPOC}(wd)}{\text{RRPOC}(100)} = \left(\frac{wd}{100}\right)^{-b} \tag{1}$$

where RRPOC(100) is the flux at 100 m and the attenuation coefficient of $b = 0.86$ is considered to be a globally averaged value (we shall return to this point later). We fit this power function using an analytical model for POC advection and degradation in the water column in order to derive the apparent reactivity of organic matter raining to the seafloor, $k_{app}(wd)$. Three POC reactivity classes, $i$ (termed highly reactive, reactive, and refractory), and two particle size classes, $j$ (small and large) characterized by different sinking speeds were included. POC was degraded using first-order kinetics. We considered

that three reactivity classes is the minimum required to be able to simulate the Martin curve by summing a series of exponential functions.

The degradation of each POC fraction in the water column was defined as:

$$-\frac{d\text{POC}_{i,j}(wd)}{d\,wd} \cdot w_j = k_{i,j} \cdot \text{POC}_{i,j}(wd) \tag{2}$$

here $k_{i,j}$ represents the first order degradation constants. The model was solved for $wd \geq 100$ m with the boundary condition:

$$\text{POC}_{j,i}(100) = \text{POC}_{\text{tot}}(100) \cdot f_j \cdot f_i \tag{3}$$



with $POC_{j,i}(100)$ being the concentration of each size/reactivity class at 100 m water depth, $POC_{tot}(100)$ is the total POC concentration at 100 m, and $f_j$ and $f_i$ represent the size and reactivity class fractions, respectively. The contribution of large POC particles to total POC at 100 m water depth was set to 20 % ($f_{large} = 0.2$) based on global observations and fits to tracer distributions in the ocean, including iron, phosphate, silicate, nitrate and ammonium concentrations as well as benthic $O_2$

uptake using the global biogeochemical model NEMO-PISCES (Aumont et al., 2017). Large and small particles were assigned a sinking speed of 50 m d$^{-1}$ and 2 m d$^{-1}$, respectively, based on the same model. The nominal cut-off size between these two size classes is assumed to be 100 μm (Aumont et al., 2015). These simplifications are necessary, considering that sinking speeds range over several orders of magnitude from $10^1$ to $10^3$ m d$^{-1}$ (Kriest and Oschlies, 2008). Moreover, particles can undergo a variety of aggregation, disaggregation and repackaging process during settling. The uncertainties created by

these additional factors will be discussed later.

In the global model (Aumont et al., 2017), the degradation rate $k$ was the same for both size classes of POC and depended on temperature. In order to reduce the number of adjustable parameters, we made the rather strong assumption that the attenuation of small and large particles was the same. Therefore, the rate constants of small particles were corrected to the ratio of sinking speeds:

$$k_{small,i} = k_{large,i} \cdot w_{small,i}/w_{large,i} \tag{4}$$

The solution of Eq. 2 has the following form:

$$POC_{j,i}(wd) = exp\left((100 - wd)\frac{k_{j,i}}{w_{j,i}}\right) POC_{tot}(100) \cdot f_j \cdot f_i \tag{5}$$

$$POC_{tot}(100) = \sum_{j,i} POC_{j,i}(100) \tag{6}$$

The apparent reactivity, $k_{app}(wd)$, is then equal to:

$$k_{app}(wd) = \frac{\sum_{j,i} k_{j,i} POC_{j,i}(wd)}{\sum_{j,i} POC_{j,i}(wd)} \tag{7}$$

With the assumption put forward in Eq. (4), the sinking speed of small particles does not affect the POC concentration profile. However, $w_{small}$, or more correctly ratio $w_{small} / w_{large}$ does directly impact the rate of POC degradation and $k_{app}$.

The model (Eq. 5) was fit to the Martin curve for water depths ≥100 m by adjusting the three decay constants ($k_{large,highly\_reactive}$, $k_{large,reactive}$ and $k_{large,refractory}$) and the relative contribution of the two reactive POC fractions ($f_{reactive}$ and

$f_{refractory}$). The reactive fraction was calculated as $1 - f_{highly\_reactive} - f_{refractory}$. The five unknown parameter values were constrained using Monte-Carlo type analysis by varying $k_{large,highly\_reactive}$ from 50 to 200 yr$^{-1}$, $k_{large,reactive}$ from 5 to 50 yr$^{-1}$, $k_{large,refractory}$ from 0.1 to 5 yr$^{-1}$. These ranges match values determined for fresh plankton detritus sinking through the water column (McDonnell et al., 2015). The value for $f_{highly\_reactive}$ was varied from 0.1 to 0.8 and $f_{refractory}$ from 0.1 to 0.4, based on observations that the bulk of fresh POC is labile (Arndt et al., 2013; Belcher et al., 2016). A total of 127 combinations of

values for $k_{j,i}$ and $f_i$ that resulted in a fit of POC($wd$) to Eq. 1 to within $R^2$=0.98 (the cloud of grey lines in Fig. 1) were then averaged to determine $k_{app}(wd)$ (solid line in Fig. 1). The continuous decrease in $k_{app}$ with depth reflects the gradual





consumption of labile organic matter moieties in the sinking material and the increasing proportion of refractory material (Wakeham et al., 1997; Dauwe et al., 1999).

It should be noted that we attempted to carry out the above procedure with one particle size only. However, the ability of the model to simulate the Martin curve was much reduced and very sensitive to the prescribed sinking speed, resulting in a wide range of $k_{app}$. The use of two particle types with different sinking speeds is therefore justified and required to provide more robust estimates of $k_{app}$.

As a first approximation, the simulated $k_{app}$ can be approximated with the following power function:

$$k_{app}(wd) = 3731 \cdot wd^{-1.011} \tag{8}$$

The modelled $k_{app}$ then serves as the basis for determining the rate constants of the sediment model, as described in the following section.

## 2.2 Multi-G model of organic matter degradation in the sediment

In this section we describe how the rate of POC degradation in bioturbated surface sediments, $RPOC(x,t)$, is calculated as a function of sediment depth, $x$. The rate of POC degradation at the sediment-water interface, $RPOC(0,t)$, is assumed to be continuous with POC degradation in the water column (Arndt et al., 2013). Hence, $k_{app}(wd)$ provides the sediment model with the surface value of apparent reactivity of the POC raining to the seafloor. As for the water column, POC degradation in sediments (in dry weight percent (wt.%) $yr^{-1}$) is based on the decay of discrete POC reactivity fractions, $i$. The POC degradation rate at zero depth can be described as:

$$RPOC(0,t) = k_{app}(wd) \cdot POC_{tot}(0,t) = \sum_i k_i \cdot POC_i(0,t) \tag{9}$$

where $POC_{tot}(0,t)$ (in wt.%) is the total POC content at the sediment-water interface and $k_{app}$ (in $yr^{-1}$) is the apparent reactivity of total POC pool derived in the previous section. Thus, $k_{app}$ at the sediment-water interface is continuous with $k_{app}$ in the water column. In sediments, $POC_{tot}(0,t)$ is equally assumed to be represented by three similar fractions as in the water column: highly reactive, reactive and refractory but with no distinction of size fractions. The depth-dependent rate of POC degradation through the bioturbated zone can thus be described as follows:

$$RPOC(x,t) = \sum_i k_i \cdot POC_i(x,t) = k_0 \cdot POC_0(x,t) + k_1 \cdot POC_1(x,t) + k_2 \cdot POC_2(x,t) \tag{10}$$

where $k_{0,1,2}$ (in $yr^{-1}$) represent first order rate constants and $POC_{0,1,2}$ denotes organic matter content of highly reactive, reactive and refractory POC pools, respectively. Note that in keeping with previous definitions, $k_0$ refers to the portion of POC that is degraded at the sediment water interface (Boudreau, 1997; Arndt et al., 2013). Degradation rate constants characterizing these fractions differ from those in water column. This is because organic material that becomes buried and mixed into the sediment column becomes subject to a multitude of preservation effects induced by mineralogical interaction that reduce its overall bioavailability (Hedges and Keil, 1995). As a result, the reactivity of organic matter continuously decreases with sediment depth (i.e. time, Middelburg, 1989). In this approach, the first order constants are constant although the bulk reactivity does decrease with depth in the sediment in line with the diminishing contributions of the more labile



fractions. The reactivity of the individual POC fractions undergoing degradation in the sediment (Eq. 10) is achieved based on the known apparent reactivity of material raining to the seafloor by varying the relative abundances of each reactive class:

$$
\begin{cases}
k_0 \cdot f_0 + k_1 \cdot f_1 + k_2 \cdot f_2 = k_{app} \\
f_0 + f_1 + f_2 = 1 \\
f_1 = fr \cdot f_0 \\
f_2 = CBE
\end{cases}
\tag{11}
$$

where $f_0$, $f_1$ and $f_2$ are the dimensionless fractions of highly reactive, reactive and refractory POC at the sediment-water interface. The refractory fraction $f_2$ is considered to be largely buried below bioturbated zone and so equal to the carbon burial efficiency, CBE (Eq. 20 below). This is the fraction of the POC deposited on the seafloor that is buried below the bioturbated layer that does not communicate with the ocean on ~$10^3$ yr timescales. From the outset, therefore, the model is consistent with global rates of POC burial, essentially leaving two reactive fractions to simulate diagenesis within the bioturbated layer.

In Eq. (11), $k_0$, $k_1$ and $k_2$ are parameters to be constrained. To solve the equation, $f_0$ and $f_1$ are linked through the dimensionless parameter $fr$, which can be defined as a function of $k_{app}$ (Fig. 1) and CBE after solving Eq. 11:

$$
fr = \frac{-CBE \cdot k_0 + k_0 + CBE \cdot k_2 - k_{app}}{CBE \cdot k_1 - k_1 - CBE \cdot k_2 + k_{app}}
\tag{12}
$$

Obviously, $fr$ should be positive, which is the case when the numerator and denominator of Eq. 12 are positive (if they are both negative, then $fr$ is still positive, but $k_0 < k_1$):

$$
-CBE \cdot k_0 + k_0 + CBE \cdot k_2 - k_{app} > 0
\tag{13}
$$

$$
CBE \cdot k_1 - k_1 - CBE \cdot k_2 + k_{app} > 0
$$

If these two inequalities are solved for $k_0$ and $k_1$ respectively, one obtains:

$$
k_0 > \frac{k_{app} - CBE \cdot k_2}{1 - CBE} > k_1
\tag{14}
$$

If the prescribed values of discrete reactive classes are considered to be constant, $k_0$ should be greater, and $k_1$ lower, than the maximum and minimum value of the solution from Eq. 14, respectively. Considering a CBE for accumulating fine-grained shelf sediments at continental margins of 40-60% (Burdige, 2007) an upper end value of $k_0$ of 70 yr$^{-1}$ can be derived (Fig. 1, dashed line). The minimum value of $k_{app}$ is about 0.8 y$^{-1}$ at 4000 m water depth. Extrapolating $k_{app}$ to greater water depths suggests a value of $k_1$ of 0.5 yr$^{-1}$. The $k_0$ value fits within the range of POC mineralization constants in sinking material in the water column, which have been reported to be as high as 180 yr$^{-1}$ (McDonnell et al., 2015).

The third refractory POC class is assumed to be largely buried below the bioturbated zone and characterized by low non-zero reactivity of 0.001y$^{-1}$. This value is taken from the range of values the least reactive fractions in 3-G model studies of surface sediments from around the world compiled by Arndt et al. (2013). However, for the deep sea where sedimentation rate is very low, the residence time of POC in bioturbated zone is long enough to allow considerable degradation of refractory fraction. To avoid this, $k_2$ was scaled to the sedimentation rate:

$$
k_2 = 0.001 \cdot \omega_{acc}(wd) / \omega_{acc}(0)
\tag{15}
$$



where $\omega_{acc}(wd)$ is the sedimentation rate at a given water depth and $\omega_{acc}(0)$ is the sedimentation rate at zero water depth calculated using an empirical function based on water depth (Burwicz et al., 2011). The corresponding lifetimes ($1/k$) of reactive classes are ~1 week, 2 yr and 1000 yr. The sensitivity of the model to these constants will be explored later.

Finally, the derived rate constants were used to simulate the degradation and concentration of POC in bioturbated surface sediments (upper 10 cm) using a 1-D reaction transport model following the classical multi-G approach (Jørgensen, 1978, Berner, 1980). The reactions in the model are similar to those described in Stolpovsky et al. (2015). In that model, solutes also included are $O_2$, $NO_3^-$, $NO_2^-$, $NH_4^+$ and oxygen-demand units (ODU) that represent reduced products of anaerobic organic carbon mineralization (i.e. sulfide, dissolved iron and manganese). The benthic fluxes of $O_2$ and $NO_3^-$ are simulated and compared to a global databased of $n = 185$ observations. Porewater distributions in the porewaters are further used for

ground truthing the model, although far fewer data are available.

POC in the model is transported through the sediment by accumulation considering compaction and mixing by bioturbation. Solutes were transported by advection, molecular diffusion, and bioirrigation (Stolpovsky et al., 2015). Partial differential equations were used to solve the concentration changes with time until a steady state was reached. For each POC fraction (in wt.%), the relevant equation is:

$$\left(1-\phi(x)\right)\frac{\partial POC_i(x,t)}{\partial t} = \frac{\partial\left(\left(1-\phi(x)\right)\cdot D_B(x)\frac{\partial POC_i(x,t)}{\partial x}\right)}{\partial x} - \frac{\partial\left(\left(1-\phi(x)\right)\cdot v_{solids}(x)\cdot POC_i(x,t)\right)}{\partial x} - \left(1-\phi(x)\right)k_i\cdot POC_i(x,t) \qquad (16)$$

where $t$ (yr) is time, $x$ (cm) is depth below the sediment-water interface, $\phi$ (dimensionless) is porosity, $D_B$ (cm$^2$ yr$^{-1}$) is the bioturbation coefficient, and $v_{solids}$ (cm yr$^{-1}$) is the solid burial velocity.

For solutes ($C_i(x)$ in mmol cm$^{-3}$ of pore fluid):

$$\phi(x)\frac{\partial C_i(x,t)}{\partial t} = \frac{\partial\left(\phi(x)D_{S-i}(x)\frac{\partial C_i(x,t)}{\partial x}\right)}{\partial x} - \frac{\partial(\phi(x)v_{solutes}(x)C_i(x,t))}{\partial x} + \phi(x)\alpha_i\left(C_i(0) - C_i(x,t)\right) + \Sigma\phi(x)R_i(x,t) \qquad (17)$$

where $D_{S-i}$ (cm$^2$ yr$^{-1}$) is the tortuosity-corrected molecular diffusion coefficient of species $i$, $v_{solutes}$ (cm yr$^{-1}$) is the solute burial velocity, $\alpha_i$ (yr$^{-1}$) is the bioirrigation coefficient, and $\Sigma R_i$ is the sum of biogeochemical reactions affecting $C_i$. Constitutive equations for $\phi$, $D_B$, $D_S$, $v_{solutes}$, $v_{solutes}$ and $\alpha$ are given in Stolpovsky et al. (2015).

The reaction network includes the major reactions of oxygen and nitrogen cycles in surface sediments. Mineralization of POC is linked to aerobic respiration, nitrate and nitrite reduction, and anaerobic respiration producing ODUs. The electron

acceptors are used sequentially in the order $O_2$, $NO_2^-$ and $NO_3^-$ using hyperbolic kinetics and appropriate limiting constants. The reaction network as well as model parameters are listed in Tables 1 and 2 in Stolpovsky et al. (2015).

The unit conversion factor between POC (wt. %) and solutes (mmol cm$^{-3}$) due to chemical reactions is:

$$n = \frac{\phi \cdot 12 \text{ g C (mol C)}^{-1}}{\rho \cdot (1-\phi(x))\cdot 10} \qquad (18)$$

where $\rho$ (g cm$^{-3}$) is the dry sediment density.

Fixed concentrations were imposed for solutes (Dirichlet boundary) at the sediment surface ($x = 0$ cm). Measured bottom water concentrations were used for $O_2$, $NO_3^-$ and $NH_4^+$ whereas $NO_2^-$ and ODU were set to zero since they do not accumulate





in seawater to a significant degree. The upper boundary condition of each POC fraction was defined as a fraction of the rain rate:

$$RRPOC_i = f_i \cdot RRPOC \qquad (19)$$

RRPOC at each station in the database was imposed as a constant value derived from the depth-integrated rate of POC

degradation in the bioturbated layer, $RPOC_B$ (mmol m$^{-2}$ d$^{-1}$), and the CBE after solving the following system of equations:

$$\begin{cases} RRPOC - RRPOC \cdot CBE = RPOC_B \\ CBE = 0.013 + 0.53 \cdot \dfrac{RRPOC^2}{(7.0+RRPOC)^2} \end{cases} \qquad (20)$$

where CBE is calculated according to an empirical function that depends on rain rate (Dunne et al., 2007). $RPOC_B$ was approximated from a mass balance of the measured benthic fluxes of $O_2$, $NO_3^-$ and $NH_4^+$ (Stolpovsky et al., 2015).

At the bottom of the bioturbated layer (x = 10 cm), a zero gradient (Neumann) boundary was applied for all species.

**3 Results**

The rain rate of POC to the seafloor for each site in the database is compared with the Martin curve ($b$ = 0.86) in Fig. 2a. The good agreement between the two demonstrates that the independent rain rate estimates are largely consistent with the theoretical flux of sinking material based on open-ocean sediment trap observations (Martin et al., 1987). With increasing water depth, the fraction of reactive POC, characterized by $k_1$, increases at the expense of the highly reactive fraction, $k_0$

(Fig. 2b). The scaling factor $fr$ increases from 0 at the shallowest depths to around 250 in the deep sea, meaning that < 1 % of reactive detritus is highly reactive by the time it reaches great ocean depths. This is caused by the more rapid mineralization of labile components in sinking organic detritus.

Simulated $O_2$ fluxes for the global database fit very well to measured data using the imposed $k_0$ and $k_1$ of 70 yr$^{-1}$ and 0.5 yr$^{-1}$, respectively (red symbols, Fig. 3a). Since $O_2$ is the ultimate electron acceptor, either for the direct oxidation of POC or for

oxidation of the reducing equivalents of anaerobic respiration (i.e. ODUs), this goodness-of-fit demonstrates that the total carbon degradation rate is correctly simulated at each station. The $NO_3^-$ result shows more scatter (Fig. 3b). The benthic N cycle is convoluted with multiple sources and sinks and with vertical stratification of processes caused by redox sensitivities of the corresponding reactions (Table 1 in Stolpovsky et al., 2015). The reaction rate constants of the N cycle are unknown at the global scale. At some stations, the model strongly underestimates the observed $NO_3^-$ flux. For example, those inside the

green ellipse correspond to the Peruvian margin where $NO_3^-$ uptake is dominated by active transport by nitrate-storing sulphur oxidizing bacteria such as *Thioploca* spp. (Fossing et al., 1995; Dale et al., 2016). The current model does not consider this process.

Both $O_2$ and $NO_3^-$ fluxes are not particularly sensitive to 10-20 % variations in $k_0$ and $k_1$ (coloured symbols on plots). Although the $NO_3^-$ fluxes are more sensitive than $O_2$, no combination of these parameters is able to improve a fit at all the

sites simultaneously, which can be put down to structural deficiencies in the model. Even so, the measured fluxes are reproducible with the model to within ca. 20 % over the prescribed variations in the rate constants. We take this to mean that





the $k_{app}$ provides a robust constraint on the mineralization constants and the benthic fluxes. It is clear that the fluxes would be more sensitive to e.g. very low rate constants, but this would violate the $k_{app}$ constraint imposed by the Martin curve.

As a further validation of the approach, modeled and measured vertical geochemical profiles were compared for the same dataset as used to verify the power law model (Stolpovsky et al., 2015). These sites include diverse settings, such as the

continental shelf (41 and 114 m water depth), upper slope (241 and 1025 m) and deep-sea (3073 m) sediments (Fig. 4). Two of the sites are located in high-nitrate-low-oxygen (HNLO) areas where poorly oxygenated waters bathe the seafloor (Washington and Mauritanian margin). The 3-G model (red curves) captures the observed trends in $O_2$ profiles through the bioturbated layer at these sites and also reproduces the trends in $NO_3^-$ porewater concentrations. The prominent subsurface peaks in modeled $O_2$ and $NO_3^-$ at St. NH14A and St. H are apparently absent in the field data and probably caused by too

intense bioirrigation in the model. The rate of this process is described using an empirical function based on total oxygen uptake and shows considerable uncertainty when applied globally (Meile and Van Cappellen, 2003). Nevertheless, these peaks do not affect $O_2$ and $NO_3^-$ gradients at the sediment surface, and the benthic fluxes are simulated accurately.

As a test of our assumption that three reactive size classes are required for simulating the data, we can compare our result with a 2-G model that excludes one of the reactive fractions (i.e. $f_r = 0$, Eq. (11)). In this case, the rate constant for the single

reactive fraction, $k*$, can be obtained from the following equation:

$$(1 - \mathrm{CBE}) \cdot k* + \mathrm{CBE} \cdot k_2 = mk \qquad (21)$$

where $k_2$ is defined in the same way as in 3-G model. Values of $k*$ are thus intermediate between $k_0$ and $k_1$. The benthic fluxes derived with the 2-G model compare well with those derived with the 3-G model, although higher-end $NO_3^-$ fluxes tend to be overestimated relative to the 3-G model (Fig. 5). However, the porewater profiles using the 2-G model are very

poorly simulated, especially for $NO_3^-$, due to the lack of carbon mineralization below the oxic layer where most of N cycling takes place. We conclude that the 2-G model is structurally inferior to the 3-G model for simulating N sources and sinks accurately.

An alternative way to visualize this result is to compare the simulated $NO_3^-$ penetration depth (NPD) for an additional set of 12 stations that was used to validate the 1-D diagenetic model 'Muds' (Archer et al., 2002). NPD is defined as the sediment

depth where $NO_3^-$ concentration falls to 2% of the local bottom water level. The NPD is a useful comparative metric because the depth-profile of organic matter degradation largely determines the depth where $NO_3^-$ is depleted. The 3-G model is able to predict the NPD at 10 from 12 stations (exceptions being #16 and #32) to within a factor of two or 1.5 cm, whereas for the 2-G model this decreases to 5 stations (Fig. 6). This again demonstrates that one effective reactive POC pool defined by a single rate constant is insufficient to represent the reactivity of natural organic matter. At a minimum, two reactive fractions

undergoing degradation on the time-scale of burial through the bioturbated layer, plus a poorly reactive fraction that is largely buried, are required to simulate the vertical structure of $O_2$ and $NO_3^-$ diagenesis in addition to POC burial.





## 4 Discussion

### 4.1 Sedimentary organic carbon degradation constants

As recently pointed out by Keil (2017), the mechanisms and processes that determine the flux of organic carbon in the ocean interior (e.g., the biological pump) and preservation within sediments are among today's great oceanographic challenges

(Doney and Karnauskas 2014; Heinze et al. 2015). Indeed, inclusion of organic carbon burial in sediments improves model distributions of redox sensitive tracers in the ocean such as oxygen and nutrients (Palastanga et al., 2011; Kriest and Oschlies, 2013) and, consequently, predictions of future climate scenarios. Proper parameterization of organic carbon degradation kinetics in sediments is thus of utmost importance for the forthcoming generation of ESMs that explicitly recycle and/or preserve sinking planktonic detritus in the sediment compartment.

In this study, we derived a 3-G diagenetic model that is able to simulate a global database of benthic $O_2$ and $NO_3^-$ fluxes and that is further consistent with global POC burial rates (based on CBE) and porewater $O_2$ and $NO_3^-$ distributions. The motivation for this work is to provide a methodology for parametrizing POC degradation in ESMs of the global carbon cycle. In our approach, the corresponding rate constants of the highly reactive ($k_0$ = 70 yr$^{-1}$), reactive ($k_1$ = 0.5 yr$^{-1}$) and poorly reactive classes ($k_2$ = 0.001 yr$^{-1}$) are fixed. However, the relative proportion of these classes varies globally as a

function of the apparent reactivity of sinking POC, $k_{app}$. To apply this model in a global model, one needs to know $k_{app}$ (Eq. 7) and the buried fraction, CBE. The latter is readily calculated from the empirical function of Dunne et al. (2007) given in Eq. (20). The 'classic' Martin curve ($b$ = 0.86), whilst not without its detractors (see below), seems to provide a robust constraint on a 3-G reactivity model of organic matter being degraded in the sediment, at least to a first approximation. The model can nonetheless be applied in ESMs that do not use the Martin curve as long as the $k_{app}$ of POC sinking to the seafloor

can be estimated. The 3-G model performs as good as our previous simulations of the same database using a continuous power law description of POC degradation (Stolpovsky et al., 2015). These workers showed that even through the bioturbated layer the apparent rate constant for POC mineralization can decrease by orders of magnitude, and varies from site to site.

The rate constants for $k_0$ and $k_1$ derived here are suitable for diagenesis in the surface sediment that is of most interest to the

global modelling community. Our proposed value of the highly reactive fraction $k_0$, equivalent to a lifetime (1/k) of around 1 week, is toward the high end of those reported previously for $k_0$ using 'stand-alone' diagenetic models. In a review on this topic, Arndt et al. (2013) report that values of $k_0$ in 3-G diagenetic models tuned to field data show a distinct maximum of 10$^1$ yr$^{-1}$. The highest value reported was 76 yr$^{-1}$, derived for an Arctic fjord using a carefully constrained empirical model, for a POC fraction that is degraded just below the sediment-water interface (Berg et al., 2003). One can thus argue that our $k_0$ is

reasonable, even when applied to the deep-sea. Direct mineralization measurements of sinking POC suggest apparent first-order rate constants of bulk POC as high as 180 yr$^{-1}$ (McDonnell et al., 2015). Furthermore, certain components of biological pump, such a mineral ballasting or diatom coagulation, can rapidly deliver fresh organic matter to great ocean depths (Buesseler et al., 2008; Belcher et al., 2016). Our findings nonetheless differ with the result of a well-documented two year-



long phytoplankton decomposition experiment by Westrich and Berner (1984). They extracted a 3-G mineralization model for POC from their data with rate constants of $24 \pm 4$, $1.4 \pm 0.7$ and $0$ yr$^{-1}$. Whilst other incubation studies cited by these workers provide very similar values, we were unable to simulate the $NO_3^-$ fluxes or porewater profiles in our database using their rate constants. This may be because the sampling resolution in the experimental study was insufficient to capture the

degradation of POC with very short lifetimes. In fact, our rate constants are able to simulate the temporal decrease in phytoplankton carbon with the same degree of accuracy as those derived by Westrich and Berner despite the large differences in POC attenuation in the initial stages of the experiment (Fig. 7; $R^2 = 0.98$ and standard error = 0.06 g/l in both cases).

It is noteworthy that our $k_0$ value, and those listed in Arndt et al. (2013), tend to be orders of magnitude higher than those

used in ESMs that explicitly couple sediment and pelagic POC dynamics. For instance, rate constants for aerobic respiration in the 1-D benthic model coupled to HAMOCC range from 0.005 to 0.01 yr$^{-1}$ (Palastanga et al., 2011), whilst benthic organic matter mineralization in the MEDUSA model is associated with a turnover of 0.024 yr$^{-1}$ (Munhoven, 2007). Sedimentary organic matter mineralization constants in ESMs are typically tuned to global organic carbon burial rates or sediment POC content (see Arndt et al., 2013). In contrast, 'stand-alone' diagenetic models are usually confirmed against a suite of site-

specific porewater concentration profiles. These models are more likely to reveal the presence of POC that degrades at the sediment-water interface compared to global biogeochemical models tuned to fit sedimentary POC burial or content. Indeed, the high $k_0$ value of 76 yr$^{-1}$ mentioned above was determined using $O_2$ concentrations measured by microelectrodes in the surface millimetres (Berg et al., 2003). Limitations on diagenetic models, however, often imposed by coarse sampling resolution, may not capture vertical gradients in solute concentrations at the sediment–water interface (which only provide

information on net rates) resulting from the degradation of the $k_0$ pool. By contrast, less reactive fractions will degrade over larger sediment depths and their mineralization rate can be more easily constrained by e.g. accumulation of ammonium in the porewater (Berner, 1980). In situ benthic flux measurements of highly reactive species such as $O_2$ and $NO_3^-$ integrate the degradation of all POC classes. In combination with modelling, in situ fluxes constitute a powerful constraint on the mineralization of the highly reactive POC fractions and associated turnover of $O_2$ and $NO_3^-$ (Bohlen et al., 2011; Dale et al.,

2016). This may partly explain why our $k_0$ value is at the higher end of those in the model database assembled by Arndt et al. (2013), most of which were not constrained by in situ flux measurements.

## 4.2 Upscaling and uncertainties

The multi-G model for benthic POC mineralization proposed here compartmentalizes organic matter into well packaged macromolecular fractions that obey first-order kinetics with rate constants that are fixed and independent of processes taking

place in the sediment. This is, of course, a gross oversimplification of reality. Sedimentary organic matter is a complex array of chemical components characterized by large divergences in degradabilities that are further altered by mineral interactions, geopolymerization, thermodynamic factors and the activity of micro-and macro-biota (e.g. de Leeuw and Largeau, 1993; Aller, 1994; Mayer, 1995; LaRowe and Van Cappellen, 2011). These, plus other unknown and poorly understood factors



(Burdige, 2007), lead to a continuous decrease in POC degradation over time, that is, with an apparent time-dependent rate constant (Middelburg et al., 1989). We are still far from a mechanistic understanding of the fundamental controls on organic matter degradation dynamics in diverse marine settings. Explicit formulations to capture the structural complexity of nature are thus currently beyond the capacity of diagenetic models, and the multi-G approach remains the cornerstone of most

diagenetic models published to date (Arndt et al., 2013). Clearly, the major controls on organic matter degradation and preservation can be encapsulated within the multi-G concept with considerable success.

This observation implies that conceptually simple approaches like the multi-G model can be coupled to global biogeochemical models or ESMs to account for sediment feedbacks in simulations of climate-relevant processes. The need for benthic-pelagic coupling in ESMs has driven the search for a generalized and transferable framework for predicting

sedimentary POC degradation kinetics at the regional and global scale. Yet, a review of the literature (>250 data sets) by Arndt et al. (2013), building upon previous work (Toth and Lerman, 1977; Emerson, 1985; Tromp et al., 1995; Boudreau, 1997; Boudreau and Ruddick, 1991), failed to unearth any statistically robust relationship between the rate constants for organic matter degradation (both multi-G and continuum models) and major controlling factors including RRPOC, sedimentation rate and water depth. In fact, it would appear that the rate constants $k_0$, $k_1$ and $k_2$ are completely independent of

these drivers (Arndt et al., 2013). This is consistent with our findings, where the k values are fixed. Instead, we propose that an important consideration is the changing *proportion* of the relative reactive classes reaching the seafloor in addition to their reactivity. The partitioning of sinking material into discrete reactive classes is linked to the apparent reactivity of the material itself, which we approximated using the Martin curve. Thus, the degradation of benthic organic carbon in our model is consistent with, and set by, processes occurring in the water column. With this constraint, global trends in sedimentary POC

degradation become apparent when discrete organic matter reactivity classes and their relative abundances are both treated as adjustable and independent variables.

Inclusion of highly reactive POC fractions in diagenetic models imposes limits on the structural complexity of vertically-resolved benthic models coupled to global biogeochemical models. Diagenetic models are typically solved over hundreds of vertical sediment layers whose thickness increases from e.g. sub-mm scale at the sediment surface to e.g. several centimetres

at dm or m in deeper strata. By comparison, and mainly due to computational constraints, benthic models coupled to ESMs tend to have a relatively low number of layers (e.g. 11 layers over the upper 10 cm in HAMOCC; Palastanga et al., 2011). Consider a 1-D advection-diffusion-reaction problem for the first-order decay of POC ($k$ = 70 yr$^{-1}$) in the bioturbated layer (L = 10 cm) on the lower continental slope (2000 m). For these settings, $D_B$ = 0.5 cm$^2$ yr$^{-1}$ and $\omega_{acc}$ = 0.005 cm yr$^{-1}$ (see references in Stolpovsky et al., 2015). The Peclet number is < 1 for these conditions, meaning that diffusive transport

(bioturbation) dominates advective transport (burial). Further, for highly reactive POC, the characteristic time scale for POC degradation is much smaller than for POC transport. To maintain model accuracy, therefore, the spatial discretization in the diagenetic model ($\Delta x$) must then be smaller than the relevant length scale, $(D/k)^{0.5}$, which in this example is less than 1 mm. This limitation is maintained throughout the ocean for typical values of $D_B$ and $\omega_{acc}$, requiring a stable routine for solving the diagenetic model. With a lower k value of 0.01 yr$^{-1}$ as in the ESM mentioned above, the maximum $\Delta x$ would increase



significantly. These calculations are approximate since the dynamics of redox sensitive elements is typically non-linear, and here we only intend to demonstrate the need for higher spatial resolution as $k$ increases to capture the fine-scaled distributions of sedimentary variables. This is an important consideration for many processes, particularly for carbonate dissolution by respiratory $CO_2$ which takes place in the upper millimetres of the sediment (Hales, 2003; Jahnke and Jahnke,

2004). In general, for highly reactive classes, large reductions in the spatial resolution of the diagenetic model will lead to inaccurate benthic feedbacks for redox sensitive elements such as $O_2$, $NO_3^-$, and $PO_4^{3-}$. Given these caveats and the computational penalties demanded by online coupling of diagenetic models, it is hardly surprising that global or regional models favour computationally efficient empirical transfer functions to simulate benthic feedbacks of N, P, Fe and $O_2$ cycles (Middelburg et al., 1997; Soetaert et al., 2000; Capet et al., 2016; Bohlen et al., 2012; Somes et al., 2013; Dale et al., 2015;

Wallmann, 2010; Yang and Gruber, 2016).

The application of our proposed multi-G model relies on knowledge of $k_{app}$ (Eq. 7) and the CBE (Eq. 20). Both of these are imperfectly known at the global scale. With regards to $k_{app}$, the depth attenuation of carbon flux in the ocean is critical to estimate the apparent reactivity of organic carbon raining to the seafloor. The power law (the Martin curve) model describing the POC flux through the water column has found wide application in global models (e.g. Najjar et al., 1992; Ito and

Follows, 2005). However, concerns about the accuracy of the Martin curve in capturing small-scale spatial and temporal variability of the biological pump in the upper ocean have called into question its suitability for global scale applications (Buessler et al., 2007). A rather wide range of the exponent $b$ between 0.4 and 1.75 has been reported and discussed at length (Keil, 2017; Guidi et al., 2015 and references therein). To a first approximation, however, the Martin curve with a $b = 0.86$ captures much of the RRPOC database (Fig. 2a). This supports the use of the apparent reactivity of POC derived from the

Martin relationship, even though $b$ may vary both spatially and temporally (Henson et al., 2015).

Trends in carbon burial efficiencies are generally understood throughout the ocean: low CBEs characterize the deep-sea (< 5 %) versus higher CBEs on the margins (>20 %). Yet, data are limited and subject to error due to mismatches in the time-scales of input parameters needed to determine CBE (Burdige, 2007). The mechanistic controls on CBE are still not resolved (Hedges and Keil, 1995). However, since most POC in sediments away from the margins is degraded, error in the CBE will

generally have a minor influence on our results.

Finally, an additional source of uncertainty rests with the sinking speed of particulate organic matter through the water column and the distribution of particle sizes, both of which were used to derive the apparent reactivity of settling POC (see Methods). We parameterized these according to the NEMO-PISCES model that is tuned to biogeochemical tracers in the water column as well as global distribution of oxygen fluxes at the sediment–water interface (Aumont et al., 2017). Whilst it

is beyond the scope of the present study to discuss the uncertainty in these model aspects, it worth noting that the contribution of large POC particles to total POC in NEMO-PISCES (assumed to be 20 %; Aumont et al., 2017) varies throughout the ocean. Larger particles in sinking material are associated with eastern boundary systems and areas of higher primary production, contributing up to 50 % of the total (O. Aumont, pers. comm). Since larger particles will tend to sink more quickly, the flux of POC to the seafloor in these regions will be greater than predicted by the Martin curve (i.e. $b <$





0.86). However, parameterizing the sinking velocities is evidently not straightforward since they span several orders of magnitude (meters to thousands of meters per day) (Kriest and Oschlies, 2008). Even small changes in the sinking speed or remineralization rates can change the *e*-folding depth for carbon export by hundreds of meters (DeVries et al., 2012). Ecosystem structure, and the associated packaging, disaggregation, and repackaging processes during settling may create

more spatial and temporal variability in POC sinking flux, hence its apparent reactivity, than assumed by a simple globally-applicable power law (De La Rocha and Passow, 2007; Wilson et al. 2008; Henson et al., 2012). Moreover, lateral transport of POC on the continental margins to deep-sea depocenters can be accelerated in benthic and intermediate nepheloid layers and by mass wasting events (Jahnke et al., 1990; Thomsen, 1999; Inthorn et al., 2006), further complicating our understanding of the biological pump.

**5 Conclusions**

Biogenic elements (C, N, P, Fe) that are packaged into organic material in the sunlit ocean are either mineralized in the ocean interior or deposited on the seafloor. Most organic material reaching the seafloor is mineralized and returned to the ocean. The rest is permanently lost from the bioavailable pool either by microbial activity (e.g. N loss to $N_2$ by denitrification bacteria) or through mineral precipitation and/or simple burial. Over long time-scales, these sediment sinks begin to exert a

major control on nutrient inventories, ocean fertility, and climate-ocean interactions. Realistic model simulations of ocean chemistry in past, present and future scenarios need to account for the recycling of organic material at the seafloor.

In an attempt to improve our understanding of the rate and depth of organic carbon mineralization in seafloor surface sediments, we parameterized a 1-D diagenetic model for organic carbon mineralization that can be coupled to Earth System Models. In contrast to previous approaches, the apparent reactivity of the organic material degraded in the seafloor is

continuous with, and set by, the apparent reactivity of material sinking through the water column. We propose that an important and mostly overlooked consideration in previous upscaling approaches is the *proportion* of the relative reactive classes reaching the seafloor in addition to their reactivity. The results imply the presence of a highly reactive organic carbon class at the sediment water interface. Mineralization of this pool will have important implications for the flux of redox sensitive elements to/from the ocean as well as for carbonate dissolution by respiratory $CO_2$.

Many unknowns of course remain and influence the predictive capabilities of our findings. Further studies of the dynamics of POC mineralization in the water column and its reactivity at the point of deposition on the seafloor are pressing. Until the spatial heterogeneity of the downward flux of organic matter in the ocean is more completely understood and predictable, knowledge of the reactivity of organic matter raining to the seafloor will be limited to the very few sites worldwide where benthic studies have been undertaken. Even if the flux and reactivity of POC reaching the sediment throughout the ocean

could be unambiguously determined (which they likely cannot), structural uncertainty in complex diagenetic models that account for the entire spectrum of boundary conditions encountered at the seafloor will become evident in the incredible lack of sedimentary data from the ocean basins and contrasting margin environments. Diagenetic models and ESMs alike are, by




definition, simplifications of nature and will require simplified global relationships like the one proposed here for the foreseeable future.

**Acknowledgements**

This work is a contribution of the Sonderforschungsbereich 754 "Climate-Biogeochemistry Interactions in the Tropical Ocean" (www.sfb754.de) financially supported by the Deutsche Forschungsgemeinschaft (DFG). We thank Iris Kreist and Ivan L'Heureux for helpful discussions.

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

Figure 1: Apparent reactivity ($k_{app}$, solid line) of total POC raining to the seafloor (degraded and buried fractions) obtained after averaging of apparent reactivities that resulted in a fit of POC(*wd*) to Eq. 1 (the cloud of grey lines). The effective apparent reactivity ($k_{app\_eff}$, dashed line) reflects the apparent reactivity of POC without considering the refractory buried fraction.





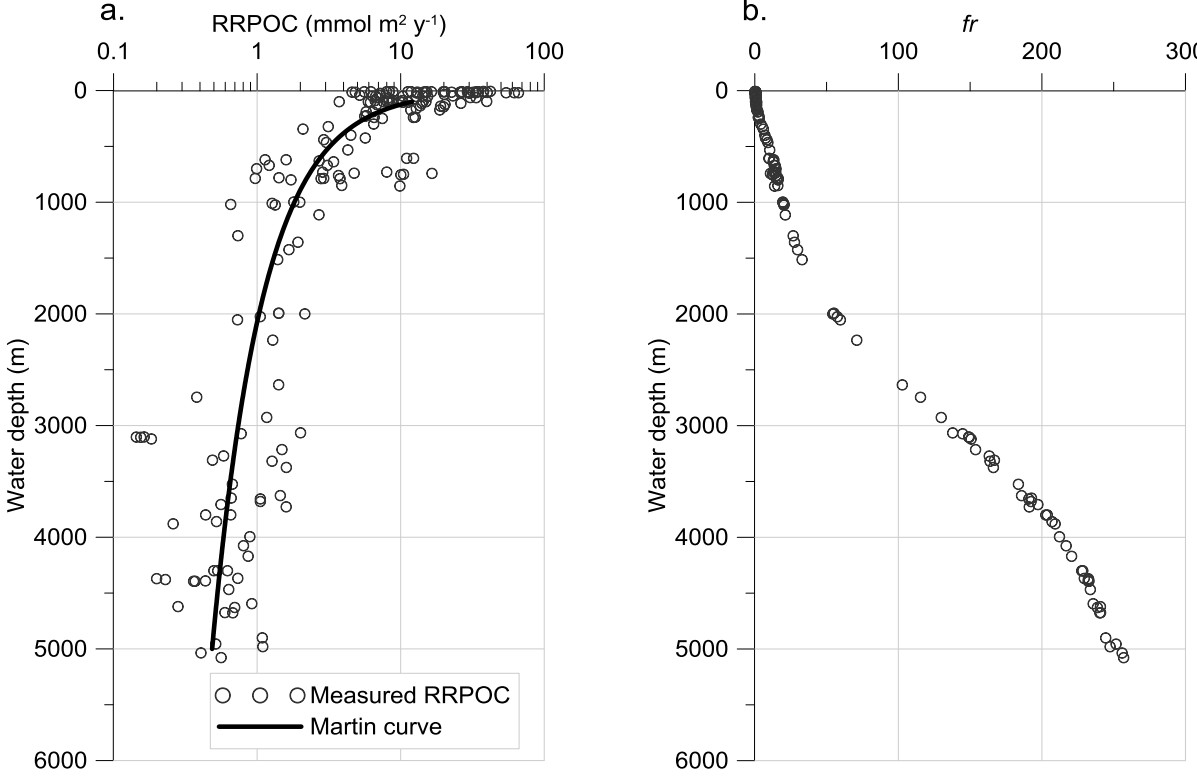

Figure 2: (a) Organic carbon rain rates (RRPOC, Eq. 20) derived for each station in the database versus the Martin curve (Eq. 1). The flux at 100 m used to derive the Martin curve is 12 mmol m$^{-2}$ d$^{-1}$ (average of 16 stations between from 90 to 114 m water depth). The data agree with a Martin coefficient of $b$ = -0.86. (b) The dimensionless parameter $fr$ (Eq. 12) describing the relative abundance of the reactive POC fraction in sediments (POC$_1$) versus the highly reactive fraction (POC$_0$). The sigmoidal distribution of the data arises from the dependency of CBE on organic carbon rain rate (Eq. 20).




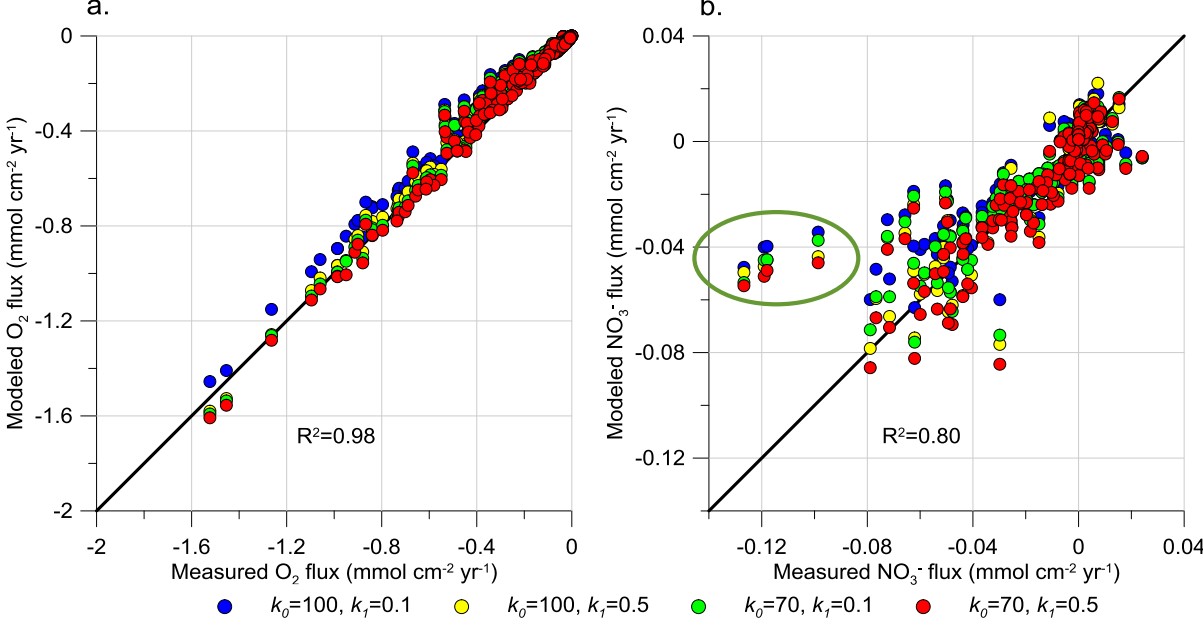

**Figure 3: Modeled versus measured benthic fluxes of (a) $O_2$ and (b) $NO_3^-$. The solid line indicates the 1:1 correlation. Different coloured symbols show the sensitivity of the fluxes for different combinations of rate constants $k_1$ and $k_2$; the red symbols denote**

5 **the baseline fit used in Fig. 4 to 6. The area marked with a green ellipse indicates the stations on the Peruvian margin characterized by enhanced $NO_3^-$ uptake by sulphur oxidizing bacteria (Dale et al., 2016). The $R^2$ values for baseline fit are provided on each plot ($p$-values are << 0.05).**



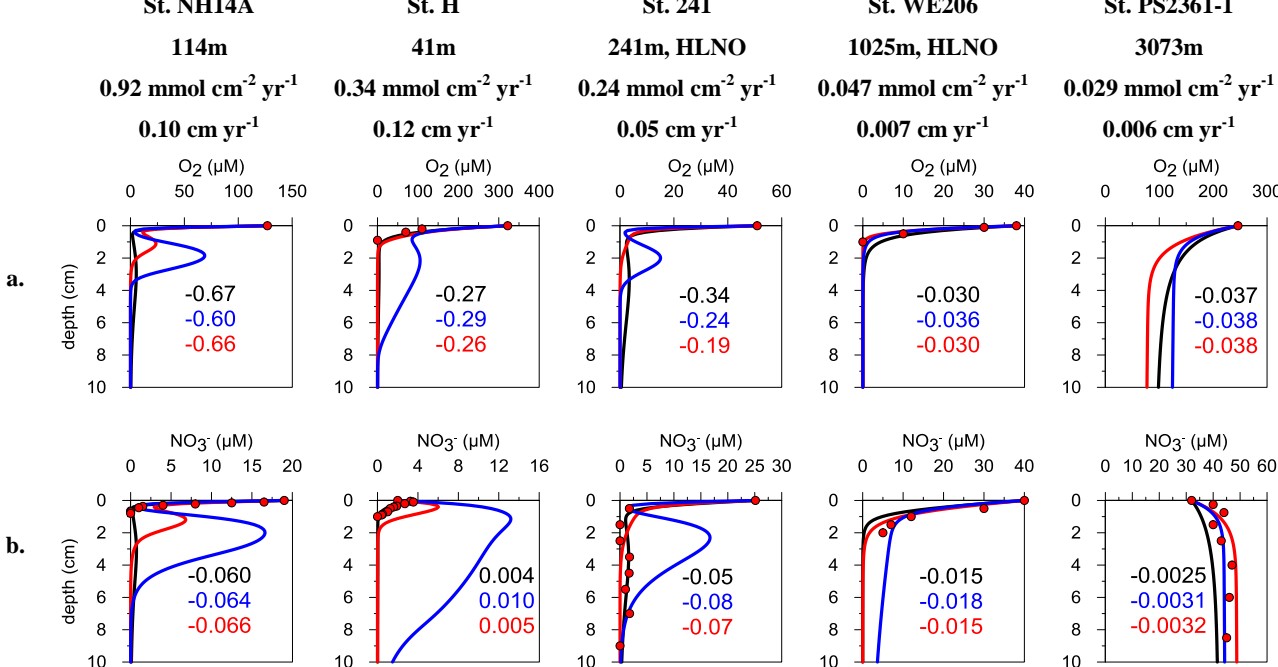

**Figure 4:** (a) $O_2$ and (b) $NO_3^-$ concentrations in the bioturbated layer for marine settings with different water depths, POC rain rates and sedimentation rates. Black curves show the model-predictions with the power function (Stolpovsky et al., 2015), red curves (this study) denote the 3-G model, blue curves show results using 2-G approach, and the symbols are the measured data. The numbers on each panel indicate measured (black), modelled with 2-G model (blue) and 3-G model (red) fluxes in mmol cm$^{-2}$ y$^{-1}$. The stations differ in their water depth, POC rain rate and sedimentation rate (listed above the plots). References for stations (see Table S1 in Supplement in Stolpovsky et al., 2015): St. NH14A = Washington margin (Devol and Christensen, 1993), St. H = Arctic shelf (Devol et al., 1997), St. 241 = Mauritanian margin (Dale et al., 2014), St. WE206 = Washington margin (Hartnett and Devol, 2003), St. PS2361-1 = Southern Ocean (Smetacek et al., 1997).



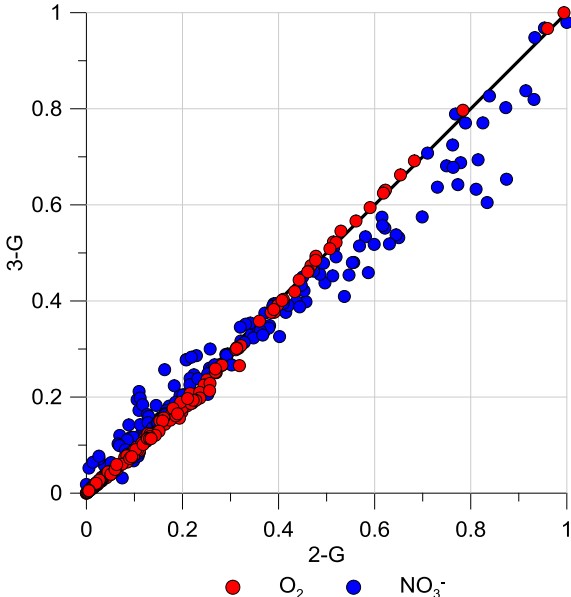

**Figure 5: Normalized benthic O$_2$ and NO$_3^-$ fluxes for the 3-G versus 2-G models. The solid line indicates the 1:1 correlation.**





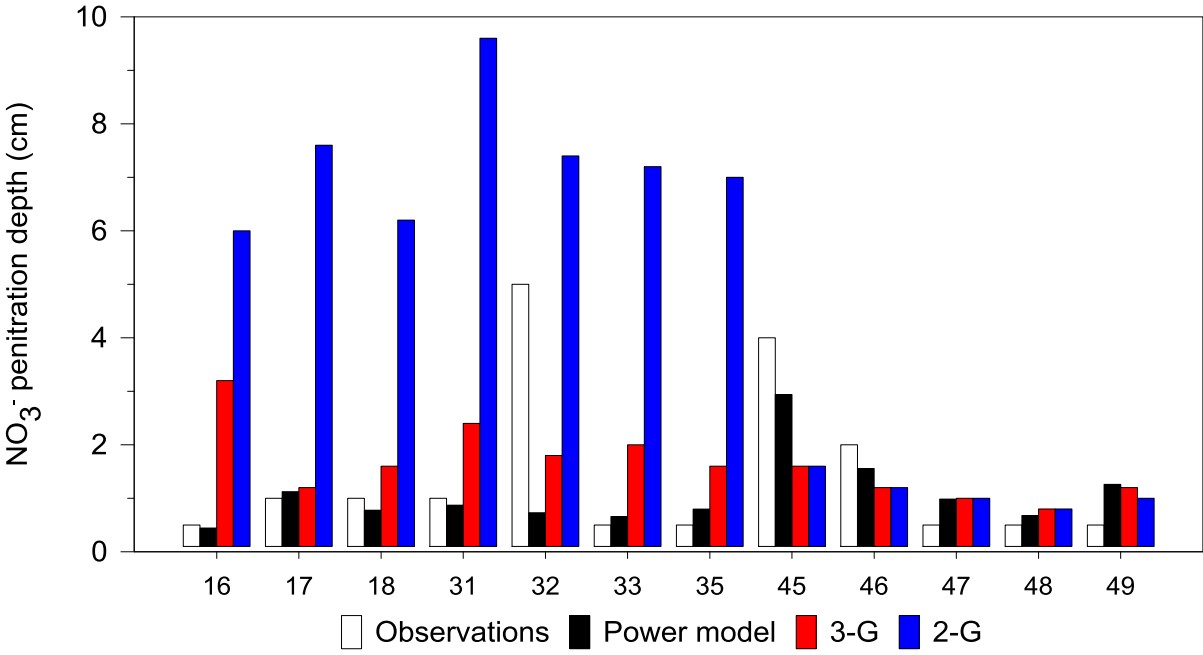

**Figure 6: Observed and simulated nitrate penetration depths (NPD in cm) for sites compiled by Archer et al. (2002) that are independent to those in our database. The numbering on the x-axis denotes the data set number given in Table 2 in Archer et al. (2002). The power model result is from Stolpovsky et al. (2015). To simulate the data, measured RRPOC and bottom water $O_2$ and**
5  **$NO_3^-$ listed in Archer et al. (2002) were used as boundary conditions.**



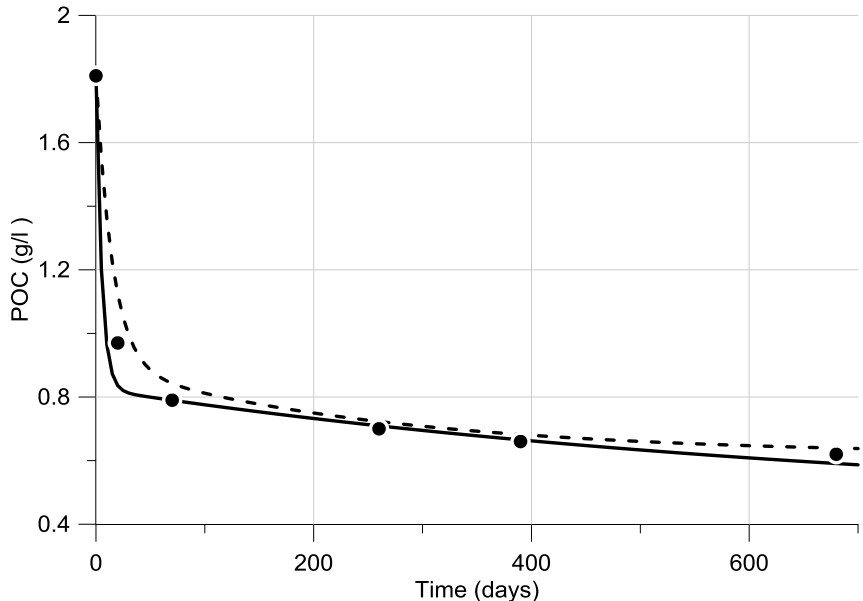

**Figure 7: Measured POC from degradation experiments with fresh phytoplankton (black circles) along with the 3-G fit from Westrich and Berner (1984) (dashed line; $k_0 = 24$ yr$^{-1}$, $k_1 = 1.4$ yr$^{-1}$, $k_2 = 0$ yr$^{-1}$, and $f_0 = 0.50$, $f_1 = 0.16$, $f_2 = 0.34$, $R^2 = 0.98$). The solid line shows the result with parameters derived in this study ($k_0 = 70$ yr$^{-1}$, $k_1 = 0.5$ yr$^{-1}$, $k_2 = 0.001$ yr$^{-1}$ and $f_0 = 0.54$, $f_1 = 0.21$, $R^2 = 0.98$).**