# Peer review of "A new look at the multi-G model for organic carbon degradation in surface marine sediments for coupled benthic-pelagic simulations of the global ocean"

_Biogeosciences, 2017_

## Referee Comment (RC1) · Anonymous Referee #1 · 2 Nov 2017

This paper presents a three-component (3-G) diagenetic model for POC remineralization in near-seafloor bioturbated sediment, where the three components are degraded over time scales between few days and 1000 years. The model parameters are constrained by a database of oxygen and nitrate fluxes at the seafloor and are also validated against measured profiles of oxygen and nitrate pore water concentrations in sediment. The goal of the work is to provide a general model of near-seafloor diagenetic processes useful for global biogeochemical Earth system models. The diagenetic model and the data that constrain it are clearly explained, the comparison of model

predictions and measurements is discussed thoroughly, and the manuscript presents a solid procedure to model POC remineralization in near-seafloor marine sediments.

I have some comments that the authors should address, however, concerning the relationship of this paper to earlier work by the same authors (Stolpovsky et al. 2015; hereafter SDW2015). The paper under discussion is based on the the diagenetic modeling procedure of SDW2015, except that it uses a 3-G formulation for POC remineralization where each of the three fractions of POC is degraded at a constant reaction rate. SDW2015 instead applied a continuum formulation where the overall reaction rate decreased with sediment age and depth. The fundamental basis for a continuously decreasing reaction rate is the recognition that POC is a mixture of components that have a broad range of reactivities. As the most reactive fractions are consumed first, the overall reaction rate will diminish during burial; see the lucid presentation of this point by Boudreau and Ruddick (1991, quoted in the paper).

A continuum formulation based on the recognition that POC has a spectrum of reactivities seems to be a better representation of reality compared to a multi-G model with a finite number of components. The question is then: in what way is the 3-G model presented in this paper an improvement over that of SDW2015? The fit to measured fluxes and concentrations seems to be good in both models, and the continuum model is also defined by a small number of parameters that describe the decrease of reaction rate during burial. The authors stress that the model in the paper under discussion is constrained by the Martin curve for POC degradation in the water column; is this an important difference with SDW2015? If so, why not update the continuum representation? Perhaps I am missing something and the 3-G model has other advantages. In my opinion, this paper needs a clear explanation of why it is a step forward compared to SDW2015 as opposed to a subsidiary modeling exercise.

I also list a number of suggestions to improve the text and figures in the following.

P3 L30: A typo in Equation 3: POC_j,i should be POC_i,j.

[Figure]

P3 L23: The POC fraction that degrades over a 1000-yr time scale is termed here "refractory." There is plenty of evidence for for POC remineralization taking place much deeper than the bioturbated zone (e.g., microbial methanogenesis beneath the sulfate reduction zone), and POC that degrades at time scales well above 1000 yr is not well described as refractory. A better term, used on P10 L14, would be "poorly reactive."

P4 L19: The "apparent reactivity" defined here is the weighted average of the three reaction rates (Equation 7), and it corresponds to the mean of a continuous distribution of POC reaction rates. A better term for this parameter would be "average reactivity" or "bulk reactivity:" the "apparent" refers to the 3-G model, which is an approximation of reality that the authors recognize (P11 L30).

P6 L11-24: I found the introduction of the parameter fr=f1/f0 very confusing. Is it necessary to get bounds on k0 and k1? If that's the case, it should not be used further in the paper, where it can be substituted by the ratio f1/f0, which is more meaningful to the reader.

P7 L2: There is no listed reference to the quoted Burvicz et al. 2011. This is an important source, as the sedimentation rates control the actual value of k2 in the modeling (Equation 15).

P7 L9: What are "Porewater distributions in the porewater?" Do you mean concentrations?

P8 L9: The zero-gradient boundary condition at the base of the bioturbated layer (10 cm) is fine for oxygen and nitrate (Figure 4), but it may not be adequate for reduced products of anaerobic POC remineralization (ODUs) such as sulfide generated by sulfate reduction beneath the bioturbated layer. On the other hand, maybe the overall contribution of ODUs is small and this is not an issue.

P11 L30: The authors recognize here the simplification of the 3-G model they use; this important issue should be stressed in the abstract and conclusions.

Figures 1 and 7: These two figures show the decrease of apparent reactivity with water depth and of POC with time. In both cases, this decrease approximately follows a power law, with values that rapidly decrease with increasing depth/time. It is hard to judge fits and differences in these plots, because they are dominated by few relatively large values near the origin. I suggest adding to these two figures a plot in log-linear or log-log scales, which will show the fit for all values.

---

## Referee Comment (RC2) · Anonymous Referee #2 · 8 Nov 2017

This manuscript focuses on modeling particulate organic matter degradation in marine sediments. The authors correctly stress that this topic needs more work in order to improve our quantitative description of benthic biogeochemical cycling on regional and global scales. An existing sub-model for organic matter degradation (the multi-G model with 3 fixed decompositions rates: 70, 0.5, 0.001 yr-1) is adopted, and most of the manuscript focuses on parameterizing the rest of the model without detailed knowledge from site-specific biogeochemical sediment measurements. The model is tested against a large global database of sediment-water fluxes and sediment profiles

for oxygen and nitrate.

I do find the work to be important, but the manuscript has a few troubling parts:

As said, the manuscript goes through a lot of considerations/effort to parameterize the proposed model. I understand the rational for this, but I was not convinced that it actually improved the accuracy of several parameter assessments. In other words, I am concerned that all of this effort based on numerous assumptions gives a false impression of how well the model is constrained in reality, and thus, how robust its predictions will be. In that light, I find that several model assumptions should be validated better. Admittedly, this is not an easy task, but it would give much more credibility to the model if it was validated in greater detail than as described. Specifically, I suggest to validate it against a couple of existing comprehensive datasets produced by detailed site-specific biogeochemical measurements. In addition to sediment water fluxes and depth profiles of all simulated species, including ammonium, it should include parameters such as the ratio between oxic/anoxic organic matter degradation. Without realistic reproductions by the model of all included variables, future model predictions and scenarios can be seriously misleading. Some comprehensive datasets are described in the Wang and Van Cappellen (1996), Rysgaard et al. (1998), and Fossing et al. (2004 - NERI Technical Report, No. 483).

Because 'losses' of reduced species, for example via burial, are usually small, oxygen uptake of the sediment should match the input of degradable organic matter well. This point is reflected here in Fig. 3a. However, a total sediment depth of only 10 cm is used in the model and with the lower Neumann boundary conditions used, the results in Fig. 3 imply that all organic matter is degraded over that depth. This is in conflict with many published studies that have measured substantial degradation activity well below 10 cm. This hints that the chosen fixed rate constants for the slower organic matter degradation – a key-part of the model – are off, at least for some sites.

The manuscript does a good job reviewing earlier modeling work that is similar. However, it is unclear to me how and why this new model stands out relative to those. Why is this model application better? What are the improvements and advantages? These distinctions should be made.

The manuscript is fairly well organized, but due to a dense writing style, the numerous details given, and many abbreviations use, it is not the easiest one to read. This could be lighten up, by tabulating some of the information and by placing some of the detailed derivatives in one or two appendices.

Finally, it is stated that one key assumption is that the "rate of POC degradation at the sediment-water interface, RPOC(0,t), is assumed to be continuous with POC degradation in the water column (Arndt et al., 2013)". In other words, the rate of POC degradation in the bottom of water column is used to parameterize the upper sediment POC degradation rate. This assumption seems to me to be a stretch given the substantial differences in microbial community compositions and densities between the bottom water and in top of the sediment. I may be wrong here, but I do not recall that the Arndt paper supports this crucial assumption.

If the authors can address the above concern the remaining revisions to the paper should be minor. Below are listed some specific comments:

Page 1, line 17: This seems to be in conflict with the statement starting on page 10, line 33.

Page 2, line 4: I don't know the policies of BD, but to give 9 references to support a statement seems to be too much. Some journals will not allow more than three references.

Page 2, line 18: Define in few words the 'reactive continuum model'.

Page 4, line 12: What is meant by 'a rather strong assumption'?

Page 7, line 7: I believe that the clever definition in terms of diagenetic modeling of an ODU pool was first proposed by Soetaert et al. (1996). This should be recognized.

Page 7, line 12: Solutes can be transported by advection, molecular diffusion, irrigation, and also by 'diffusion-like' bioturbation. The latter can be as important as molecular diffusion (Aller and Aller, 1992).

Page 8, line 20: Am I misunderstanding something here? With the total input of degradable organic matter know and with the model boundary conditions used, the good match between the measured and simulated oxygen flux is expected. It does not mean that it "demonstrates that the total carbon degradation rate is correctly simulated at each station".

Page 12, line 25: A highly effective numerical solution to this prohibitive modeling problem was proposed by Berg et al. (2007) and should be cited.

---

## Referee Comment (RC3) · Anonymous Referee #3 · 8 Nov 2017

The manuscript describes an early diagenetic model in which the mineralization of POC is formulated using 3 POC fractions which differ in their reactivity. This is a commonly used approach embracing minimum complexity while still being able to reproduce flux or porewater observations. The main emphasis of the paper is to establish a parameterization of such a model that is applicable across a wide range of conditions, yet is simple enough to be possibly lend itself for use in Earth System Models. The model achieves this by only requiring knowledge of the burial efficiency and the apparent reactivity of the organic matter at the sediment water interface. These two parameters

are determined from a published relationship of the burial efficiency with the rain rate, and by establishing a connection between kapp and to the settling organic matter via the Martin curve.

The presentation is quite clear and concise, and the authors nicely reiterate the main points in the discussion and conclusion. I also appreciate the significant effort put into the comparison of model simulations to observational data, and the discussion of uncertainties. The latter addresses some of the concerns that I had reading through their methodology (see below). However, there are a number of assumptions I found surprising and that may warrant more discussion.

- use of the Martin curve: this is a good starting point, and the concerns regarding the use of this relationship are discussed. However, it is unclear how sensitive the results are to the value of b (the paper makes the case that b=0.86 matches the data but it doesn't address the sensitivity to this parameter and papers such as the one cited by Buesseler et al. illustrate the wide range of values for b) - the burial efficiency relationship reported in Dunne et al. is used to estimate the proportion of the refractory POC pool and to constrain the rate constants. This estimate of the CBE is applied at 10 cm, presumably well above the depth at which mineralization becomes negligible (which clashes with the choice of no gradient lower boundary conditions at 10cm for POC and mineralization products). Furthermore, the CBE estimate of Dunne et al. is off by about 80% (the average of (estimated-true burial eff.)/true burial efficiency for the data shown in Figure 2 of Dunne et al.). A central benefit of the work presented here relates to its applicability across a wide range of oceanic conditions. I suggest that the authors quantify and show the impact of the considerable uncertainties in the fundamental input variables b and CBE.

- particle classes (page 4): As the authors state, the choice of two particle size classes with distinct sinking velocities is a tremendous simplification (see e.g. Jackson et al. 1997, DSR I 44: 1739-67). Furthermore, the assumption is then made that the ratio of the rate constants for small and large particles of the same POC pool i match the ratio

of the settling velocities (equation 4). I simply don't understand the rationale for this particular choice (e.g., if the POC was the same in the small and large fraction of pool i, then one may expect the rate constants to be the same, resulting in a more pronounced removal of slow settling small particles, which is not what is implemented here). If there is compelling evidence for such a scaling of rate constants, then this needs to be presented, or the relevant literature needs to be cited. Else, the consequences of such an assumption need to be quantified, or the assumption revisited.

- The rate of POC degradation is said to be continuous between sediment and water (page 5, line 14). This presumably is not true for volumetric rates, as POC concentrations change drastically at this interface. On line 20, it then says that kapp is continuous across the sediment-water interface. However, first order rate constants intrinsically reflect the abundance and activity of microbes involved in the breakdown of organic matter, which surely varies across this interface. What data is there to support this approach? (relevant because kapp is used as an anchor for the sediment reaction rate constants).

- model comparison: the 3G model is parameterized to essentially match the OC mineralization integrated over the top 10 cm (input from Martin, burial from Dunne), while parameters were optimized in the power function approach of Stolpovsky et al. 2015 to match the fluxes. Hence, it is not a big surprise that the O2 fluxes match closely, though it is not immediately evident what motivates the use of a 3G approach over the continuum approach the authors presented previously. And what about the 2G model? Can the reason it performs worse be linked to the parameterization (eq. 21; what is mk? summation over k?), in which the less reactive fraction of the 2G model is set to be equal to the refractory pool in the 3G model?

- benthic exchange fluxes reported in Figure 4 seem to be similar for profiles with similar gradients at the sediment-water interface but differ vastly below that (some have subsurface maxima). Are irrigation fluxes not important for the overall exchange, even at the shallower stations?

- Equation (2): what does w_j represent?

---

## Author Comment (AC1) · 30 Jan 2018

» A continuum formulation based on the recognition that POC has a spectrum of reactivities seems to be a better representation of reality compared to a multi-G model with a finite number of components. The question is then: in what way is the 3-G model presented in this paper an improvement over that of SDW2015? The authors stress that the model in the paper under discussion is constrained by the Martin curve for POC degradation in the water column; is this an important difference with SDW2015? If so, why not update the continuum representation?

[Figure]

We thank the review for his/her constructive comments on our manuscript. A continuum formulation is more appropriate under steady state conditions, i.e. when changes in POC rain rate (RRPOC) are slow compared to transport and reaction rates in sediments. POC degradation in the continuum model (SDW2015) is a function of RRPOC, such that RPOC over the whole sediment profile responds instantaneously to changes in RRPOC. The major advantage of G-type models is that they can be used at the short time-scales, for example temporal variations in rain rate. The temporal storage of POC or solutes is therefore explicit in the multi-G approach. We omitted this information from the final version of the paper (for length consideration) but now recognize its relevance and will include it in the revised manuscript (Reviewer #2 made a similar comment).

Specific comments

» P3 L30: A typo in Equation 3: POC_j,i should be POC_i,j.

Corrected as suggested.

» P3 L23: The POC fraction that degrades over a 1000-yr time scale is termed here "refractory." There is plenty of evidence for POC remineralization taking place much deeper than the bioturbated zone (e.g., microbial methanogenesis beneath the sulfate reduction zone), and POC that degrades at time scales well above 1000 yr is not well described as refractory. A better term, used on P10 L14, would be "poorly reactive."

We agree that POC remineralization does not end at the base of our modeled sediments (20 cm). However, for our purposes the k3 value in present study is low enough to call this fraction "refractory". We will clarify this in the revised manuscript.

» P4 L19: The "apparent reactivity" defined here is the weighted average of the three reaction rates (Equation 7), and it corresponds to the mean of a continuous distribution of POC reaction rates. A better term for this parameter would be "average reactivity" or "bulk reactivity:" the "apparent" refers to the 3-G model, which is an approximation of reality that the authors recognize (P11 L30).

We used "mean reactivity" in the older versions of the manuscript but later changed it to "apparent reactivity" to be consistent with previous studies (e.g. Arndt et al., 2013).

» P6 L11-24: I found the introduction of the parameter $fr=f1/f0$ very confusing. Is it necessary to get bounds on k0 and k1? If that's the case, it should not be used further in the paper, where it can be substituted by the ratio f1/f0, which is more meaningful to the reader.

Agreed. We will replace fr with f0/f1.

» P7 L2: There is no listed reference to the quoted Burwicz et al. 2011. This is an important source, as the sedimentation rates control the actual value of k2 in the modeling (Equation 15). Corrected.

» P7 L9: What are "Porewater distributions in the porewater?" Do you mean concentrations?

Corrected. 'in the porewaters' will be deleted.

» P8 L9: The zero-gradient boundary condition at the base of the bioturbated layer (10 cm) is fine for oxygen and nitrate (Figure 4), but it may not be adequate for reduced products of anaerobic POC remineralization (ODUs) such as sulfide generated by sulfate reduction beneath the bioturbated layer. On the other hand, maybe the overall contribution of ODUs is small and this is not an issue.

We note that the sediments were simulated to 20 cm, not 10 cm, and only the upper 10 cm was shown in Fig. 4 (will be clarified in the manuscript). Even so, anaerobic organic carbon mineralization below 20 cm will lead to non-zero concentration gradients for ODU and NH4+ at 20 cm. However, the overall contribution of anaerobic degradation below 20 cm is a small fraction of degradation in the upper 20 cm. For instance, log-log relationships of radio-tracer measurements of sulfate reduction versus depth in Aarhus Bay show that over 95% of OM is mineralized in the upper 20 cm of sediment (Holmkvist et al., 2011, GCA 75, 3581-3599). Imposing a zero-gradient condition for

ODU and NH4+ is hence justifiable. We will also stress in the revised manuscript that the kinetics of POC mineralization in deeper anaerobic sediments is not the aim of the study since it is hardly constrainable from the empirical data presented in the paper.

» P11 L30: The authors recognize here the simplification of the 3-G model they use; this important issue should be stressed in the abstract and conclusions.

We agree. Corresponding changes will be made.

» Figures 1 and 7: These two figures show the decrease of apparent reactivity with water depth and of POC with time. In both cases, this decrease approximately follows a power law, with values that rapidly decrease with increasing depth/time. It is hard to judge fits and differences in these plots, because they are dominated by few relatively large values near the origin. I suggest adding to these two figures a plot in log-linear or log-log scales, which will show the fit for all values.

Log scale gives a false feeling of importance of variation in kapp value at the deep sea (see the answer to the first question to Rev. 3), hence we prefer to keep it linear.

Please also note the supplement to this comment:
https://www.biogeosciences-discuss.net/bg-2017-397/bg-2017-397-AC1-supplement.pdf

---

## Author Comment (AC2) · 30 Jan 2018

» The manuscript goes through a lot of considerations/effort to parameterize the proposed model. I understand the rational for this, but I was not convinced that it actually improved the accuracy of several parameter assessments. In other words, I am concerned that all of this effort based on numerous assumptions gives a false impression of how well the model is constrained in reality, and thus, how robust its predictions will be. In that light, I find that several model assumptions should be validated better. Admittedly, this is not an easy task, but it would give much more credibility to the model if

it was validated in greater detail than as described. Specifically, I suggest to validate it against a couple of existing comprehensive datasets produced by detailed site-specific biogeochemical measurements. In addition to sediment water fluxes and depth profiles of all simulated species, including ammonium, it should include parameters such as the ratio between oxic/anoxic organic matter degradation. Without realistic reproductions by the model of all included variables, future model predictions and scenarios can be seriously misleading. Some comprehensive datasets are described in the Wang and Van Cappellen (1996), Rysgaard et al. (1998), and Fossing et al. (2004 - NERI Technical Report, No. 483).

We thank the review for his/her constructive comments on our manuscript. Regards the point made above, we would first reiterate that the model has been quasi-validated against a global database of benthic fluxes from 185 stations around the globe. This, in our view, demonstrates that the POC degradation kinetics are adequately described. The opportunity to further ground-truth the model using fluxes of other solutes is limited by the severe lack of fluxes collected in situ. Porewater data and nitrate penetration depths were used as further comparisons. We do not claim that the model can perfectly simulate field data from site-specific locations. This is obvious from Figs. 3 and 4 and arises from the use of empirical coefficients (e.g. bioirrigation, bioturbation, CBE) that themselves carry considerable uncertainties. Further validation of the model against the shelf sites proposed by the reviewed would require more complex reaction network than in our simple model, including coupled Fe-Mn-S cycling. Our model is mainly tuned to the N cycle in the upper 20 cm of the sediment, and it is here that we have most confidence in our results. In our previous study, we also made the point that the model/approach is not well suited to sites on the continental margin due to the broad heterogeneity found there. In the revised version of the manuscript we emphasize these points more clearly.

» Because 'losses' of reduced species, for example via burial, are usually small, oxygen uptake of the sediment should match the input of degradable organic matter well. This

point is reflected here in Fig. 3a. However, a total sediment depth of only 10 cm is used in the model and with the lower Neumann boundary conditions used, the results in Fig. 3 imply that all organic matter is degraded over that depth. This is in conflict with many published studies that have measured substantial degradation activity well below 10 cm. This hints that the chosen fixed rate constants for the slower organic matter degradation – a key-part of the model – are off, at least for some sites.

Firstly, all the organic is not degraded: we have three fractions and the least reactive one is set to CBE; itself independently imposed according to the function provided by Dunne et al. We agree that the lower boundary condition in the model is uncertain. Repeating the comment to Reviewer #1, we note that the sediments were simulated to 20 cm, not 10 cm, and only the upper 10 cm was shown in Fig. 4 (will be clarified in the manuscript). Even so, anaerobic organic carbon mineralization below 20 cm will lead to non-zero concentration gradients at 20 cm. However, the overall contribution anaerobic degradation below 20 cm is a small fraction of degradation in the upper 20 cm. For instance, log-log relationships of radio-tracer measurements of sulfate reduction versus depth in Aarhus Bay show that over 95% of OM is mineralized in the upper 20 cm of sediment (Holmkvist et al., 2011, GCA 75, 3581-3599). Imposing a zero-gradient condition for ODU and NH4+ is justifiable. We also make the point there and in the revised manuscript that the kinetics of POC mineralization in deeper anaerobic sediments is not the aim of the study and hardly constrainable from the empirical data presented in the paper.

» The manuscript does a good job reviewing earlier modeling work that is similar. However, it is unclear to me how and why this new model stands out relative to those. Why is this model application better? What are the improvements and advantages? These distinctions should be made.

The principal advantage is for Earth System model applications, and with temporally variable fluxes of POC to the seafloor. The over step forward is that we use the Martin curve that is water column biogeochemistry to constrain the reactivity of organic matter

reaching the seabed. This will be clarified. Please see also the response to Reviewer #1.

» The manuscript is fairly well organized, but due to a dense writing style, the numerous details given, and many abbreviations use, it is not the easiest one to read. This could be lighten up, by tabulating some of the information and by placing some of the detailed derivatives in one or two appendices.

We will include a glossary of terms in the revised manuscript.

» Finally, it is stated that one key assumption is that the "rate of POC degradation at the sediment-water interface, RPOC(0,t), is assumed to be continuous with POC degradation in the water column (Arndt et al., 2013)". In other words, the rate of POC degradation in the bottom of water column is used to parameterize the upper sediment POC degradation rate. This assumption seems to me to be a stretch given the substantial differences in microbial community compositions and densities between the bottom water and in top of the sediment. I may be wrong here, but I do not recall that the Arndt paper supports this crucial assumption.

Firstly, the Arndt reference was mis-cited and will be removed. It is true that the specific nature of sedimentary environments is vastly different from the bottom water, and other uncertainties were listed at the beginning of section 4.2. However, we take the top down view that differences in the microbial community composition and densities between the bottom water and in top of the sediment are determined by the flux of POC deposited to the sediment. In other words, we would expect that the flux of POC to the seafloor is the main factor that will determine microbial biomass and POC degradation in the seafloor. With this assumption, the reactivity of the bulk material does appear to be broadly consistent with the water column reactivity (Fig. 3 to 6). We are still at a very early stage of understanding the relationship between the continuity of POC reactivity at the sediment water interface, not least because the reactivity of material in the lower water column is probably even less well understood than in the sediments.

The assumptions made in the paper provide a basis for further research on this topic.

Specific comments

» Page 1, line 17: This seems to be in conflict with the statement starting on page 10, line 33.

There is no conflict in these two statements, and this only requires clarification. Although our derived rate constants differ with the phytoplankton decomposition experiment by Westrich and Berner (1984), they are in perfect agreement with the experimental data (Fig. 7). This is the main point we wanted to make and the text will be rephrased accordingly.

» Page 2, line 4: I don't know the policies of BD, but to give 9 references to support a statement seems to be too much. Some journals will not allow more than three references.

We will revise this.

» Page 2, line 18: Define in few words the 'reactive continuum model'.

Continuum models assume a continuous distribution of organic matter reactivity, thus avoiding the need to partition the bulk material into a defined number of discrete compound classes. A similar definition will be added to the manuscript.

» Page 4, line 12: What is meant by 'a rather strong assumption'?

We mean that there is no data to support this assumption. Please see also comment to Reviewer #3.

»Page 7, line 7: I believe that the clever definition in terms of diagenetic modeling of an ODU pool was first proposed by Soetaert et al. (1996). This should be recognized.

Correct. We will cite them appropriately in the revised manuscript.

» Page 7, line 12: Solutes can be transported by advection, molecular diffusion, irrigation, and also by 'diffusion-like' bioturbation. The latter can be as important as molecular diffusion (Aller and Aller, 1992).

Bioturbation is included in the calculated of the depth-dependent sediment diffusion coefficient: Ds = Dsw/to + DBio, where DSW = diffusion coefficient in seawater, to = tortuosity and DBio = bioturbation coefficient.

» Page 8, line 20: Am I misunderstanding something here? With the total input of degradable organic matter known and with the model boundary conditions used, the good match between the measured and simulated oxygen flux is expected. It does not mean that it "demonstrates that the total carbon degradation rate is correctly simulated at each station".

It does because the depth-integrated rate of POC degradation is partly estimated from the carbon burial efficiency (Eq. 20). Not all the POC that arrives at the seafloor is mineralized.

» Page 12, line 25: A highly effective numerical solution to this prohibitive modeling problem was proposed by Berg et al. (2007) and should be cited.

We will include this reference.

Please also note the supplement to this comment:
https://www.biogeosciences-discuss.net/bg-2017-397/bg-2017-397-AC2-supplement.pdf

---

## Author Comment (AC3) · 30 Jan 2018

» It is unclear how sensitive the results are to the value of b (the paper makes the case that b=0.86 matches the data but it doesn't address the sensitivity to this parameter and papers such as the one cited by Buesseler et al. illustrate the wide range of values for b) - the burial efficiency relationship reported in Dunne et al. is used to estimate the proportion of the refractory POC pool and to constrain the rate constants. This estimate of the CBE is applied at 10 cm, presumably well above the depth at which mineralization becomes negligible (which clashes with the choice of no gradient lower

boundary conditions at 10cm for POC and mineralization products). Furthermore, the CBE estimate of Dunne et al. is off by about 80% (the average of (estimated-true burial eff.)/true burial efficiency for the data shown in Figure 2 of Dunne et al.). A central benefit of the work presented here relates to its applicability across a wide range of oceanic conditions. I suggest that the authors quantify and show the impact of the considerable uncertainties in the fundamental input variables b and CBE.

We thank the review for his/her constructive comments on our manuscript. The sensitivity of the apparent reactivity (kapp) to a $\pm 20$ % change in b is shown on linear (Fig. 1) and log (Fig. 2) scales. More negative values of b lead to a greater attenuation (loss) of POC with water depth. This difference in apparent reactivity leads to minor (within 50% error) changes in flux, thus for station 1 from Figure 4 (114 m) the flux of oxygen calculated with b = −0.69 differs from the one calculated with b = −1.032 by 14% and nitrate by 13.5%; and for station 5 (3073 m) by 0.02% and 0.13% respectively. With regard to the CBE algorithm, we recognize that the Dunne relationship is not perfect at the global scale. However, it is broadly consistent with global modeling studies (see Fig. 1 in Kriest and Oschlies, 2013) as well as data evaluations of CBE versus rain rate by Flögel et al. (2011) and serves as a basis for correcting the POC rain rate for C burial. We agree that defining the CBE horizon is somewhat ambiguous, but we refer to the point made to Reviewer #1 and #2 that the bulk of OM is mineralized in surface sediments. Improvements could be easily made as and when new data becomes available.

Kriest, I. and Oschlies, A. (2013) Swept under the carpet: organic matter burial decreases global ocean biogeochemical model sensitivity to remineralization length scale. Biogeosciences, 10, 8401-8422. doi 10.5194/bg-10-8401-2013.

» As the authors state, the choice of two particle size classes with distinct sinking velocities is a tremendous simplification. Furthermore, the assumption is then made that the ratio of the rate constants for small and large particles of the same POC pool i match the ratio of the settling velocities (equation 4). I simply don't understand the

rationale for this particular choice (e.g., if the POC was the same in the small and large fraction of pool i, then one may expect the rate constants to be the same, resulting in a more pronounced removal of slow settling small particles, which is not what is implemented here). If there is compelling evidence for such a scaling of rate constants, then this needs to be presented, or the relevant literature needs to be cited. Else, the consequences of such an assumption need to be quantified, or the assumption revisited.

As the reviewer correctly noted, very simple representation of sinking particles (their reactivity and sinking speed) was used. The rationale for this particular choice is based on i) the assumption that small particles are settling from the very top of the water column and not originated from disintegrated large particles (aggregation/disaggregation is poorly constrained and thus not considered here) and on ii) observations proving that the contribution of POC bound in the fine fraction to the total POC concentration in the water column does not decrease with water depth (Aumont 2017, Fig. 5). To maintain this constant contribution, POC in small particles (POCsmall) must be less reactive than POC in large particles (POClarge) because the reduced sinking speed of small particles would otherwise induce a systematic decrease in the POCsmall/ POClarge ratio with water depth.

» The rate of POC degradation is said to be continuous between sediment and water (page 5, line 14). This presumably is not true for volumetric rates, as POC concentrations change drastically at this interface. On line 20, it then says that kapp is continuous across the sediment-water interface. However, first order rate constants intrinsically reflect the abundance and activity of microbes involved in the breakdown of organic matter, which surely varies across this interface. What data is there to support this approach?

Here we repeat the answer given to Reviewer #2. It is true that the specific nature of sedimentary environments is vastly different from the bottom water, and other uncertainties were listed at the beginning of section 4.2. However, the reactivity of the bulk

material does appear to be broadly consistent with the water column reactivity (Fig. 3 to 6) and it is likely that the abundance and activity of microbes is largely controlled by the flux and the reactivity of organic matter raining to the seafloor. We are still at a very early stage of understanding the relationship between the continuity of POC reactivity at the sediment water interface, not least because the reactivity of material in the lower water column is probably even less well understood than in the sediments. The assumptions made provide a basis for further research on this topic.

» The 3G model is parameterized to essentially match the OC mineralization integrated over the top 10 cm (input from Martin, burial from Dunne), while parameters were optimized in the power function approach of Stolpovsky et al. 2015 to match the fluxes. Hence, it is not a big surprise that the O2 fluxes match closely, though it is not immediately evident what motivates the use of a 3G approach over the continuum approach the authors presented previously. And what about the 2G model? Can the reason it performs worse be linked to the parameterization (eq. 21; what is mk? summation over k?), in which the less reactive fraction of the 2G model is set to be equal to the refractory pool in the 3G model?

Reviewer #1 and #2 made the same comment concerning the advantages of the 3G versus continuum model. The major advantage of G-type models is that they can be applied to non-steady state situations where temporal storage of POC and solutes in the sediment may be important. POC degradation in the continuum model is a function of RRPOC and therefore RPOC over the whole sediment profile responds instantaneously to changes in RRPOC. We omitted this information from the final version of the paper (for length consideration) but now recognize its relevance and will include it in the revised manuscript. For the 2G model, the less reactive fraction was set to be equal to the refractory pool in the 3G model in order to maintain consistency with the global CBEs. The reason why the 2G model perform worse than the 3G one is that the highly reactive fraction that is degraded over the top millimeter(s) of sediments is lacking. mk is in fact kapp; a typo that will be corrected.

» Benthic exchange fluxes reported in Figure 4 seem to be similar for profiles with similar gradients at the sediment-water interface but differ vastly below that (some have subsurface maxima). Are irrigation fluxes not important for the overall exchange, even at the shallower stations?

Indeed, bioirrigation plays an important role in solutes transport especially at the shallower stations. However, the subsurface maxima do not play important role as POC degradation rate at that depth is very low compared to the one at sediment-water interface. The benthic flux is mostly driven by oxidation of labile POC fraction within the top millimeter(s) of sediments whereas deeper down the electron acceptors are mostly reduced with the reactive POC.

» Equation (2): what does w_j represent?

wj represents the sinking speed, and will be clarified in the manuscript.

Please also note the supplement to this comment:
https://www.biogeosciences-discuss.net/bg-2017-397/bg-2017-397-AC3-supplement.pdf

[Figure]

**Fig. 1.** The sensitivity of the apparent reactivity (k_app) to a $\pm 20$ % change in b

[Figure]

**Fig. 2.** The sensitivity of the apparent reactivity (k_app on Log scale) to a $\pm 20$ % change in b

---

## Author Response (AR1)

Dear Editor,

On behalf of all co-authors I would like to thank you for the handling the manuscript.

According to your suggestion from February 16, 2018 we hereby resubmit a revised version of the manuscript entitled "A NEW LOOK AT THE MULTI-G MODEL FOR ORGANIC CARBON DEGRADATION IN SURFACE MARINE SEDIMENTS FOR COUPLED BENTHIC-PELAGIC SIMULATIONS OF THE GLOBAL OCEAN" [Paper # bg-2017-397] for publication in Biogeosciences.

Below you can find a response to the reviews (given in boldface), in which we refer to the pages in revised manuscript where we have addressed the specific comments. Please note that particular attention was given to the English to improve the readability of the text, so most of the changes were not tracked in the text.

With kind regards,

Konstantin Stolpovsky.

**Reviewer 1**

1. A continuum formulation based on the recognition that POC has a spectrum of reactivities seems to be a better representation of reality compared to a multi-G model with a finite number of components. The question is then: in what way is the 3-G model presented in this paper an improvement over that of SDW2015? The authors stress that the model in the paper under discussion is constrained by the Martin curve for POC degradation in the water column; is this an important difference with SDW2015? If so, why not update the continuum representation?

**We thank the review for his/her constructive comments on our manuscript.**
**This question is now answered in introduction on page 3 and 4 (see the marked area with the reference to this comment).**

**Specific comments**

2. P3 L30: A typo in Equation 3: POC_j,i should be POC_i,j.

**Corrected on page 4.**

3. P3 L23: The POC fraction that degrades over a 1000-yr time scale is termed here "refractory." There is plenty of evidence for POC remineralization taking place much deeper than the bioturbated zone (e.g., microbial methanogenesis beneath the sulfate reduction zone), and POC that degrades at time scales well above 1000 yr is not well described as refractory. A better term, used on P10 L14, would be "poorly reactive."

**This issue is now discussed at page 7.**

4. P4 L19: The "apparent reactivity" defined here is the weighted average of the three reaction rates (Equation 7), and it corresponds to the mean of a continuous distribution of POC reaction rates. A better term for this parameter would be "average reactivity" or "bulk reactivity:" the "apparent" refers to the 3-G model, which is an approximation of reality that the authors recognize (P11 L30).

**We used "mean reactivity" in the older versions of the manuscript but later changed it to "apparent reactivity" to be consistent with previous studies (e.g. Arndt et al., 2013).**

5. P6 L11-24: I found the introduction of the parameter fr=f1/f0 very confusing. Is it necessary to get bounds on k0 and k1? If that's the case, it should not be used further in the paper, where it can be substituted by the ratio f1/f0, which is more meaningful to the reader.

**We replaced $fr$ with the ratio $f_1/f_0$.**

6. P7 L2: There is no listed reference to the quoted Burwicz et al. 2011. This is an important source, as the sedimentation rates control the actual value of k2 in the modeling (Equation 15).

**Corrected on page 18.**

7. P7 L9: What are "Porewater distributions in the porewater?" Do you mean concentrations?

**Corrected. 'in the porewaters' is deleted. Page 8.**

8. P8 L9: The zero-gradient boundary condition at the base of the bioturbated layer (10 cm) is fine for oxygen and nitrate (Figure 4), but it may not be adequate for reduced products of anaerobic POC remineralization (ODUs) such as sulfide generated by sulfate reduction beneath the bioturbated layer. On the other hand, maybe the overall contribution of ODUs is small and this is not an issue.

**We note that the sediments were simulated to 20 cm, not 10 cm, and only the upper 10 cm was shown in Fig. 4. It is now clarified. The question about non-zero gradient at 20cm boundary is now discussed on page 9.**

9. P11 L30: The authors recognize here the simplification of the 3-G model they use; this important issue should be stressed in the abstract and conclusions.

**We agree. Corresponding changes are made in the abstract on Page 1. This issue is also addressed in conclusion on page 16.**

10. Figures 1 and 7: These two figures show the decrease of apparent reactivity with water depth and of POC with time. In both cases, this decrease approximately follows a power law, with values that rapidly decrease with increasing depth/time. It is hard to judge fits and differences in these plots, because they are dominated by few relatively large values near the origin. I suggest adding to these two figures a plot in log-linear or log-log scales, which will show the fit for all values.

**Log scale gives a false feeling of importance of variation in $k_{app}$ value at the deep sea (see the answer to the first question to Rev. 3), hence we prefer to keep it linear.**

**Reviewer 2**

1. The manuscript goes through a lot of considerations/effort to parameterize the proposed model. I understand the rational for this, but I was not convinced that it actually improved the accuracy of several parameter assessments. In other words, I am concerned that all of this effort based on numerous assumptions gives a false impression of how well the model is constrained in reality, and thus, how robust its predictions will be. In that light, I find that several model assumptions should be validated better. Admittedly, this is not an easy task, but it would give much more credibility to the model if it was validated in greater detail than as described. Specifically, I suggest to validate it against a couple of existing comprehensive datasets produced by detailed site-specific biogeochemical measurements. In addition to sediment water fluxes and depth profiles of all simulated species, including ammonium, it should include parameters such as the ratio between oxic/anoxic organic matter degradation. Without realistic reproductions by the model of all included variables, future model predictions and scenarios can be seriously misleading. Some comprehensive datasets are described in the Wang and Van Cappellen (1996), Rysgaard et al. (1998), and Fossing et al. (2004 - NERI Technical Report, No. 483).

**We thank the review for his/her constructive comments on our manuscript.**
**The comment explaining the limited level of validation presented in the paper is given on pages 11-12 (see the marked area with the reference to this comment).**

2. Because 'losses' of reduced species, for example via burial, are usually small, oxygen uptake of the sediment should match the input of degradable organic matter well. This point is reflected here in Fig. 3a. However, a total sediment depth of only 10 cm is used in the model and with the lower Neumann boundary conditions used, the results in Fig. 3 imply that all organic matter is degraded over that depth. This is in conflict with many published studies that have measured substantial degradation activity well below 10 cm. This hints that the chosen fixed rate constants for the slower organic matter degradation – a key-part of the model – are off, at least for some sites.

**This question is now discussed on page 9.**

3. The manuscript does a good job reviewing earlier modeling work that is similar. However, it is unclear to me how and why this new model stands out relative to those. Why is this model application better? What are the improvements and advantages? These distinctions should be made.

**This question is now answered in introduction on page 3 and 4.**

4. The manuscript is fairly well organized, but due to a dense writing style, the numerous details given, and many abbreviations use, it is not the easiest one to read. This could be lighten up, by tabulating some of the information and by placing some of the detailed derivatives in one or two appendices.

**The glossary of terms is now included on page 2.**

5. Finally, it is stated that one key assumption is that the "rate of POC degradation at the sediment-water interface, RPOC(0,t), is assumed to be continuous with POC degradation in the water column (Arndt et al., 2013)". In other words, the rate of POC degradation in the bottom of water column is used to parameterize the upper sediment POC degradation rate. This assumption seems to me to be a stretch given the substantial differences in microbial community compositions and densities between the bottom water and in top of the sediment. I may be wrong here, but I do not recall that the Arndt paper supports this crucial assumption.

**The Arndt reference was mis-cited and now removed. The question about continuity between pelagic and benthic POC reactivity is now discussed on page 6.**

**Specific comments**

6. Page 1, line 17: This seems to be in conflict with the statement starting on page 10, line 33.

**There is no conflict in these two statements, and this only requires clarification. It is now clarified in abstract on page 1 and on pages 12-13.**

7. Page 2, line 4: I don't know the policies of BD, but to give 9 references to support a statement seems to be too much. Some journals will not allow more than three references.

**We find all these references to be important so we prefer to keep them all if the journal allows to.**

8. Page 2, line 18: Define in few words the 'reactive continuum model'.

**It is now defined in introduction on page 3.**

9. Page 4, line 12: What is meant by 'a rather strong assumption'?

**We mean that there is no data to support this assumption. Please see also explanation on page 5.**

10. Page 7, line 7: I believe that the clever definition in terms of diagenetic modeling of an ODU pool was first proposed by Soetaert et al. (1996). This should be recognized.

**This study is now cited, see page 8.**

11. Page 7, line 12: Solutes can be transported by advection, molecular diffusion, irrigation, and also by 'diffusion-like' bioturbation. The latter can be as important as molecular diffusion (Aller and Aller, 1992).

**Bioturbation is included in the calculated of the depth-dependent sediment diffusion coefficient. It is now clarified on page 8.**

12. Page 8, line 20: Am I misunderstanding something here? With the total input of degradable organic matter known and with the model boundary conditions used, the good match between the measured and simulated oxygen flux is expected. It does not mean that it "demonstrates that the total carbon degradation rate is correctly simulated at each station".

**It does because not all the POC that arrives at the seafloor is mineralized. This issue is now addressed on page 10.**

13. Page 12, line 25: A highly effective numerical solution to this prohibitive modeling problem was proposed by Berg et al. (2007) and should be cited.

**This reference is now included on page 14.**

**Reviewer 3**

1. It is unclear how sensitive the results are to the value of b (the paper makes the case that b=0.86 matches the data but it doesn't address the sensitivity to this parameter and papers such as the one cited by Buesseler et al. illustrate the wide range of values for b) - the burial efficiency relationship reported in Dunne et al. is used to estimate the proportion of the refractory POC pool and to constrain the rate constants. This estimate of the CBE is applied at 10 cm, presumably well above the depth at which mineralization becomes negligible (which clashes with the choice of no gradient lower boundary conditions at 10cm for POC and mineralization products). Furthermore, the CBE estimate of Dunne et al. is off by about 80% (the average of (estimated-true burial eff.)/true burial efficiency for the data shown in Figure 2 of Dunne et al.). A central benefit of the work presented here relates to its applicability across a wide range of oceanic conditions. I suggest that the authors quantify and show the impact of the considerable uncertainties in the fundamental input variables b and CBE.

**We thank the review for his/her constructive comments on our manuscript. The sensitivity of the apparent reactivity ($k_{app}$) to a ±20 % change in *b* is introduced on page 15 (see the marked area with the reference to this comment).**
**This CBE issue is also discussed on page 15.**

2. As the authors state, the choice of two particle size classes with distinct sinking velocities is a tremendous simplification. Furthermore, the assumption is then made that the ratio of the rate constants for small and large particles of the same POC pool *i* match the ratio of the settling velocities (equation 4). I simply don't understand the rationale for this particular choice (e.g., if the POC was the same in the small and large fraction of pool i, then one may expect the rate constants to be the same, resulting in a more pronounced removal of slow settling small particles, which is not what is implemented here). If there is compelling evidence for such a scaling of rate constants, then this needs to be presented, or the relevant literature needs to be cited. Else, the consequences of such an assumption need to be quantified, or the assumption revisited.

**We addressed this issue on page 5.**

3. The rate of POC degradation is said to be continuous between sediment and water (page 5, line 14). This presumably is not true for volumetric rates, as POC concentrations change drastically at this interface. On line 20, it then says that $k_{app}$ is continuous across the sediment-water interface. However, first order rate constants intrinsically reflect the abundance and activity of microbes involved in the breakdown of organic matter, which surely varies across this interface. What data is there to support this approach?

**This question is now discussed on page 6.**

4. The 3G model is parameterized to essentially match the OC mineralization integrated over the top 10 cm (input from Martin, burial from Dunne), while parameters were optimized in the power function approach of Stolpovsky et al. 2015 to match the fluxes. Hence, it is not a big surprise that the O2 fluxes match closely, though it is not immediately evident what motivates the use of a 3G approach over the continuum approach the authors presented previously.

**This question is now answered in introduction on page 4.**

5. And what about the 2G model? Can the reason it performs worse be linked to the parameterization (eq. 21; what is mk? summation over k?), in which the less reactive fraction of the 2G model is set to be equal to the refractory pool in the 3G model?

**The reason why the 2G model performs worse than the 3G one is that the highly reactive fraction that is degraded over the top millimeter(s) of sediments is lacking. Similar explanation is given on page 11. *mk* is in fact $k_{app}$; this typo in Eq. 22 (Eq. 21 in earlier version of manuscript) is now corrected.**

6. Benthic exchange fluxes reported in Figure 4 seem to be similar for profiles with similar gradients at the sediment-water interface but differ vastly below that (some have subsurface maxima). Are irrigation fluxes not important for the overall exchange, even at the shallower stations?

**This question is addressed on page 10.**

7. Equation (2): what does w_j represent?

**$w_j$ represents the sinking speed. It is now clarified in the manuscript on page 4.**

[revised manuscript text omitted]

10   to discrete compound classes with fixed reactivity (Middelburg, 1989; Boudreau and Ruddick, 1991). Implementation of continuum POC dynamics in bioturbated sediments is possible, yet cumbersome (Dale et al., 2015). Recently, Stolpovsky et al. (2015) successfully parameterized a continuum model for bioturbated sediments as a function of the POC rain rate to the seafloor using a global database of in situ flux measurements. In that model, POC degradation rate responds instantaneously to changes in rain rate and is thus applicable under quasi-steady state conditions, i.e. in settings where changes in rain rate

15   are slow compared to the mean lifetime of POC in sediments.

Benthic POC mineralization in some ESMs rely on constitutive equations that describe linear degradation kinetics of one or more sedimentary POC fractions, or 'G', as a function of e.g. burial rate (reviewed by Arndt et al., 2013). An advantage of so-called multi-G models (more than one POC class; Jørgensen, 1978) is that by modelling POC content explicitly, they can be employed in situations with marked temporal variability in POC rain rate. Whilst multi-G models work well for

20   individual sites where POC degradation can be constrained against empirical data, there is no evidence that the transferability of these models is guaranteed at the global scale. In fact, building upon previous work (Toth and Lerman, 1977; Emerson et al., 1985; Tromp et al., 1995; Boudreau, 1997; Boudreau and Ruddick, 1991), Arndt et al. (2013) showed that no statistically relationship exists between the rate constants for POC degradation and major controlling factors such as burial rate and water depth. This severely limits the predictive capacity of ESMs to faithfully reproduce, and draw

25   conclusions about, benthic feedbacks and the role of sediments in global biogeochemical cycles of carbon, nutrients and other non-conservative tracers.

In this study, we present a novel methodology to constrain the rate constants of discrete organic matter reactive classes in sediments using the multi-G approach. Our motivation is to provide a more realistic POC degradation model that can be coupled to ESMs. We employ the same global database of benthic $O_2$ and $NO_3^-$ fluxes to constrain the model that we used

30   previously to derive the continuum model (Stolpovsky et al., 2015). The model is further consistent with global POC burial rates and porewater $O_2$ and $NO_3^-$ distributions. In contrast to all previous approaches (to the best of our knowledge), the apparent reactivity of the organic material degraded in the seafloor described here is continuous with, and set by, the apparent reactivity of material sinking through the water column. This is achieved by assuming that the efficiency of carbon transfer to the ocean interior and to the seafloor (i.e. the biological pump) can be approximated using a power law, that is, the

**Comment [SK4]:** To Rev. 2, Q. 8.

**Comment [SK5]:** To Rev. 1, Q. 1.
And
To Rev. 2, Q. 3.
To Rev. 3, Q. 4.

Martin curve (Martin et al., 1987). We find that the relative proportion of the reactive classes reaching the seafloor allows for transferability, upscaling and extrapolation of the model to the global scale. The model also differs from our previous approach since it can more realistically deal with changes in carbon rain rate (Stolpovsky et al., 2015). The coherence between POC degradation in the water column and sediments should improve predictions of benthic-pelagic coupling of redox sensitive elements (e.g. $O_2$, $NO_3^-$, $PO_4^{3-}$, $Fe^{2+}$) in ESMs that explicitly account for seafloor mineralization processes.

**Comment [SK6]:** To Rev. 1, Q. 1. And To Rev. 2, Q. 3. To Rev. 3, Q. 4.

**2 Model Structure**

In this section, we first describe how the apparent reactivity ($k_{app}$ in yr$^{-1}$) of POC sinking through the water column is calculated. This information is then used to define the rate constants and fractions of three reactive POC pools in bioturbated surface sediments (upper 20 cm) by simulating global databases of $O_2$ and $NO_3^-$ fluxes using a reaction-transport sediment model. This latter model is modified from our previous study where POC degradation rates were defined using a reactive continuum approach (Stolpovsky et al., 2015).

**2.1 Apparent reactivity of organic matter sinking through the water column**

The apparent reactivity of POC raining to the seafloor was calculated by first considering the sinking flux of POC. This was achieved by simulating the Martin curve (Martin et al., 1987) that has been widely used to describe the sinking flux, or rain rate of POC to the sediment (RRPOC), as a function of water depth (*wd*, m):

$$\frac{\text{RRPOC}(wd)}{\text{RRPOC}(100)} = \left(\frac{wd}{100}\right)^{-b} \tag{1}$$

where RRPOC(100) is the flux at 100 m and the flux attenuation coefficient of $b = 0.86$ is considered to be a globally averaged value (we shall return to this point later). We fitted an analytical model for POC advection (sinking) and degradation to this power function to derive the apparent reactivity of organic matter raining to the seafloor as a function of water depth, $k_{app}(wd)$. Three POC reactivity classes, *i* (highly reactive, reactive, and poorly reactive), and two particle size classes, *j* (small and large) characterized by different sinking speeds were included. We considered that a minimum of three reactivity classes was required to be able to simulate the Martin curve accurately.

The degradation of each POC fraction in the water column was defined as:

$$\frac{d\text{POC}_{i,j}(wd)}{d\,wd} \cdot w_j = -k_{i,j} \cdot \text{POC}_{i,j}(wd) \tag{2}$$

where $k_{i,j}$ represents a first-order degradation constant and $w_j$ is the sinking speed. The model was solved for $wd \geq 100$ m with the boundary condition:

**Comment [SK7]:** To Rev. 3, Q. 7.

$$\text{POC}_{i,j}(100) = \text{POC}_{\text{tot}}(100) \cdot f_i \cdot f_j \tag{3}$$

**Comment [SK8]:** To Rev. 1, Q. 2.

where $\text{POC}_{i,j}(100)$ is the concentration of each reactivity/size class at 100 m water depth, $\text{POC}_{\text{tot}}(100)$ is the total POC concentration at 100 m, and $f_i$ and $f_j$ represent the reactivity and size class fractions, respectively. The contribution of large particles to total POC at 100 m was set to 20 % ($f_{large} = 0.2$) based on results from the global 3-D biogeochemical model

NEMO-PISCES (Aumont et al., 2017). This model was constrained against a global database of particle size distributions and dissolved tracers including iron, phosphate, silicate, nitrate and ammonium. Large and small particles were assigned a sinking speed of 50 m d$^{-1}$ and 2 m d$^{-1}$, respectively, based on the NEMO-PISCES model. The nominal cut-off size between these two size classes was assumed to be 100 μm (Aumont et al., 2015). These simplifications are necessary, considering that

5    sinking speeds range over several orders of magnitude from $10^1$ to $10^3$ m d$^{-1}$ (Kriest and Oschlies, 2008). Moreover, particles can undergo a variety of aggregation, disaggregation and repackaging process during settling. The uncertainties in our model created by these additional factors will be discussed later.

In order to reduce the number of adjustable parameters, the attenuation of small and large organic particles was the same. The rationale for this particular choice is based on i) the assumption that small particles originate at the top of the water

10    column and not from disintegration of large particles at depth, and on ii) observations that the fraction of POC bound in the fine particles to total POC concentration in the water column remains at ~ 20 % below ca. 1000 m (Aumont et al., 2017). To maintain this proportion, POC in small particles must be less reactive than POC in large particles, otherwise the reduced sinking speed of small particles would lead to a systematic increase in the $f_{large}/f_{small}$ ratio with water depth. Therefore, the mineralization constants of small particles were normalized to the ratio of sinking speeds:

15    $$k_{small,i} = k_{large,i} \cdot w_{small,i}/w_{large,i} \qquad (4)$$

With this assumption, the reduced sinking speed of small particles does not affect the POC concentration profile.

The solution of Eq. 2 has the following form:

$$POC_{i,j}(wd) = \exp\left((100 - wd)\frac{k_{i,j}}{w_{i,j}}\right) POC_{tot}(100) \cdot f_i \cdot f_j \qquad (5)$$

$$POC_{tot}(100) = \sum_{i,j} POC_{i,j}(100) \qquad (6)$$

20    The apparent reactivity, $k_{app}(wd)$, is then equal to:

$$k_{app}(wd) = \frac{\sum_{i,j} k_{i,j} POC_{i,j}(wd)}{\sum_{i,j} POC_{i,j}(wd)} \qquad (7)$$

The model (Eq. 5) was fit to the Martin curve for water depths ≥100 m by adjusting the three decay constants ($k_{large,highly\_reactive}$, $k_{large,reactive}$ and $k_{large,poorly\_reactive}$) and the relative contribution of the two reactive POC fractions ($f_{highly\_reactive}$ and $f_{poorly\_reactive}$). The reactive fraction was calculated as $1 - f_{highly\_reactive} - f_{poorly\_reactive}$. The five unknown parameter values

25    were constrained using Monte-Carlo analysis by varying $k_{large,highly\_reactive}$ from 50 to 200 yr$^{-1}$, $k_{large,reactive}$ from 5 to 50 yr$^{-1}$, $k_{large,poorly\_reactive}$ from 0.1 to 5 yr$^{-1}$. These ranges span values determined for fresh plankton detritus sinking through the water column (McDonnell et al., 2015). The values for $f_{highly\_reactive}$ and $f_{poorly\_reactive}$ were 0.1 to 0.8 and 0.1 to 0.4, respectively, based on observations that the bulk of fresh POC is labile (Arndt et al., 2013; Belcher et al., 2016). A total of 127 combinations of $k_{i,j}$ and $f_i$ that provided a fit to the Martin curve (to within a nominal $R^2$ of 0.98) is shown as the cloud of red

30    lines in Fig. 1, which were then averaged to determine $k_{app}(wd)$. The continuous decrease in $k_{app}(wd)$ with depth reflects the gradual consumption of labile organic matter moieties in the sinking material and the increasing proportion of poorly reactive material (Wakeham et al., 1997; Dauwe et al., 1999). The ability of the model to simulate the Martin curve with one

**Comment [SK9]:** To Rev. 2, Q. 9. And To Rev. 3, Q. 2.

particle size only was very much reduced, with $k_{app}(wd)$ being very sensitive to its prescribed sinking speed (result not shown). Consideration of at least two particle types with different sinking speeds seems to be required to provide robust estimates of $k_{app}(wd)$.

As a first approximation, the simulated $k_{app}(wd)$ can be approximated with the following power law (Fig. 1):

5  $$k_{app}(wd) = 3731 \cdot wd^{-1.011} \qquad (8)$$

This function is the basis for determining the rate constants of POC degradation in the sediment model, as described in the following section.

**2.2 Multi-G model of organic matter degradation in the sediment**

In this section, we describe how the kinetics of POC degradation in bioturbated surface sediments (upper 20 cm) was
10  calculated using a 1-D reaction transport model (Jørgensen, 1978, Berner, 1980). Benthic POC degradation was linked to POC reactivity in the water column. $k_{app}(wd)$ (Eq. 8) provides the sediment model with the reactivity of bulk POC at the sediment surface, from which the rate constants of discrete POC reactivity fractions, $i$ (highly reactive, reactive and refractory) were estimated. No distinction of size fractions was considered in the sediment model since all POC classes are buried and mixed through the sediment using identical transport coefficients.
15  The rate of POC degradation at the sediment surface, RPOC(0,$t$), was calculated as:

$$\mathrm{RPOC}(0,t) = k_{app}(wd) \cdot \mathrm{POC_{tot}}(0,t) = \sum_i k_i \cdot \mathrm{POC}_i(0,t) \qquad (9)$$

where $\mathrm{POC_{tot}}(0,t)$ (in dry wt.%) is the total POC content at the sediment-water interface and $k_{app}(wd)$ (in yr$^{-1}$) is its apparent reactivity, derived previously. $\mathrm{POC_{tot}}(0,t)$ is calculated directly by the model as a function of the rain rate of organic matter to the seafloor (see Eq. (19)). In this way, the mean reactivity of POC at the sediment-water interface is continuous with
20  $k_{app}(wd)$ for POC raining out of the water column for any given water depth. $k_{app}(wd)$ is simply treated as a boundary condition for the sediment model without making any mechanistic assumptions regarding, for example, microbial community structure and activity and POC preservation at the transition between the bottom water and the surface sediment. The nature of this transition is not well understood, but is assumed to be ultimately determined by the flux of POC deposited to the sediment.

25  The rate of POC degradation with depth in the sediment, $x$, is:

$$\mathrm{RPOC}(x,t) = \sum_i k_i \cdot \mathrm{POC}_i(x,t) = k_0 \cdot \mathrm{POC}_0(x,t) + k_1 \cdot \mathrm{POC}_1(x,t) + k_2 \cdot \mathrm{POC}_2(x,t) \qquad (10)$$

where $k_{0,1,2}$ (in yr$^{-1}$) represent first-order rate constants and $\mathrm{POC}_{0,1,2}$ denote highly reactive, reactive and refractory carbon, respectively. In keeping with previous studies, $k_0$ refers to highly reactive POC that is degraded at the sediment water interface (Boudreau, 1997; Arndt et al., 2013). Degradation rate constants characterizing these fractions do not have to be the
30  same as those in water column because organic material that is buried and mixed into the sediment column becomes subject to a multitude of preservation effects induced by mineralogical interactions that reduce its overall bioavailability (Hedges and Keil, 1995). Although the first-order constants in our approach are fixed, bulk POC reactivity decreases through the

**Comment [SK10]:** To Rev. 2, Q. 5.
And
To Rev. 3, Q. 3.

sediment due to diminishing contents of the more labile fractions, which approximates a continuous decrease of organic matter reactivity with sediment depth (Middelburg, 1989).

From Eq. (9), the $k_i$ values are related to $k_{app}(wd)$ via the relative abundance of each reactive class:

$$\quad \begin{cases} k_0 \cdot f_0 + k_1 \cdot f_1 + k_2 \cdot f_2 = k_{app}(wd) \\ f_0 + f_1 + f_2 = 1 \\ f_2 = CBE \end{cases} \qquad (11)$$

where $f_0$, $f_1$ and $f_2$ are the dimensionless fractions of highly reactive, reactive and refractory POC at the sediment-water interface, respectively. $f_2$ is considered to be largely buried below bioturbated zone and equal to the carbon burial efficiency, CBE (Eq. 21 below). This fraction is buried below the bioturbated layer that does not communicate with the ocean on ~$10^3$

10 yr timescales. From the outset, therefore, the model is consistent with global rates of POC burial, essentially leaving two reactive fractions to simulate diagenesis within the bioturbated layer.

The next step is to estimate $k_0$ and $k_1$ by first solving Eq. (11) for the ratio $f_1/f_0$ as a function of $k_{app}(wd)$ and CBE:

$$\frac{f_1}{f_0} = \frac{-CBE \cdot k_0 + k_0 + CBE \cdot k_2 - k_{app}(wd)}{CBE \cdot k_1 - k_1 - CBE \cdot k_2 + k_{app}(wd)} \qquad (12)$$

Obviously, $f_1/f_0$ should be positive, which is the case when the numerator and denominator of Eq. 12 are positive (if they are

15 both negative, then the ratio $f_1/f_0$ is still positive, but $k_0 < k_1$):

$$-CBE \cdot k_0 + k_0 + CBE \cdot k_2 - k_{app}(wd) > 0 \qquad (13)$$

$$CBE \cdot k_1 - k_1 - CBE \cdot k_2 + k_{app}(wd) > 0$$

Solving these two inequalities gives an expression that provides limits on $k_0$ and $k_1$ according to $k_{app}(wd)$ and CBE:

$$k_0 > \frac{k_{app}(wd) - CBE \cdot k_2}{1 - CBE} > k_1 \qquad (14)$$

20 The rate constant of the refractory POC class ($k_2 = 0.001\,\text{yr}^{-1}$) was taken from the range of values of the least reactive fractions in 3-G model studies of surface sediments from around the world (Arndt et al., 2013). Since it is largely buried below the bioturbated zone it is classed as 'refractory' for the time scales of interest here. Of course, POC remineralization does not stop at the base of the modeled sediments column, but this 'deep' POC turnover has little influence on the flux of solutes at the lower model boundary as explained below. However, for the deep sea where sedimentation is very low, the

25 residence time of POC in the bioturbated zone is long enough to allow considerable degradation of the refractory fraction. To avoid this, $k_2$ was scaled to the sedimentation rate:

$$k_2 = 0.001 \cdot \omega_{acc}(wd)/\omega_{acc}(0) \qquad (15)$$

where $\omega_{acc}(wd)$ is the sedimentation rate at a given water depth and $\omega_{acc}(0)$ is the sedimentation rate at zero water depth calculated using an empirical function (Burwicz et al., 2011).

30 The function in Eq. (14) is plotted in Fig. 1 (dashed line) alongside $k_{app}(wd)$. From the end-members of this curve, $k_0$ has to be greater than 43 yr$^{-1}$ whilst $k_1$ must be lower than $k_{app}(wd)$ in the deep sea. As baseline values in the model, we used 70 yr$^{-1}$

**Comment [SK11]:** To Rev. 1, Q. 3.

for $k_0$ and 0.5 yr$^{-1}$ for $k_1$, noting that this $k_0$ value is similar to that determined for shelf sediments based on $O_2$ microelectrode measurements (76 yr$^{-1}$, Berg et al., 2003). The corresponding lifetimes (1/$k$) of the three POC fractions are thus ~1 week, 2 yr and 1000 yr. The sensitivity of the model to these constants will be explored later. With knowledge of the rate constants and the CBE (Eq. 21), $f_1/f_0$ was calculated from Eq. (12).

5    The derived rate constants were used to simulate the degradation and content of POC (3-G model) in bioturbated surface sediments (upper 20 cm) using the 1-D reaction transport model. Solutes included $O_2$, $NO_3^-$, $NO_2^-$, and $NH_4^+$. The reaction network is similar to that described in Stolpovsky et al. (2015) and includes the major reactions of oxygen and nitrogen in surface sediments such as heterotrophic denitrification, nitrification and anammox. Mineralization of POC is linked to aerobic respiration, nitrate and nitrite reduction, and anaerobic respiration. The latter produces oxygen-demand units (ODU)

10   that lump together reduced compounds such as sulfide, dissolved iron and manganese (Soetaert et al. 1996). The reaction network and model parameters are listed in Tables 1 and 2 in Stolpovsky et al. (2015).

**Comment [SK12]:** To Rev. 2, Q. 10.

POC in the sediment model was transported by burial with compaction and mixing by bioturbation. Solutes were transported by burial, molecular diffusion, bioturbation and by the non-local transport pathway of bioirrigation that represents flushing of surface sediments by burrow-dwelling animals. Partial differential equations were used to solve the concentration changes

15   with time until a steady state was reached. For each POC fraction (in wt.%), the relevant equation is:

$$(1 - \phi(x))\frac{\partial POC_i(x,t)}{\partial t} = \frac{\partial\left((1-\phi(x))\cdot D_B(x)\frac{\partial POC_i(x,t)}{\partial x}\right)}{\partial x} - \frac{\partial\left(\omega_{acc}(wd)\cdot(1-\phi(L))\cdot POC_i(x,t)\right)}{\partial x} - (1-\phi(x))k_i \cdot POC_i(x,t) \quad (16)$$

where $t$ (yr) is time, $x$ (cm) is depth below the sediment-water interface, $\phi$ (dimensionless) is porosity, $\phi(L)$ is porosity in compacted sediments, $D_B$ (cm$^2$ yr$^{-1}$) is the bioturbation coefficient, and $\omega_{acc}(wd)$ (cm yr$^{-1}$) is the sedimentation rate as a function of water depth (Burwicz et al., 2011).

20   For solutes ($C_i(x)$ in mmol cm$^{-3}$ of pore fluid):

$$\phi(x)\frac{\partial C_i(x,t)}{\partial t} = \frac{\partial\left(\phi(x)D_{Si}(x)\frac{\partial C_i(x,t)}{\partial x}\right)}{\partial x} - \frac{\partial\left(\omega_{acc}(wd)\cdot\phi(L)\cdot C_i(x,t)\right)}{\partial x} + \phi(x)\alpha_i\left(C_i(0) - C_i(x,t)\right) + \Sigma\phi(x)R_i(x,t) \quad (17)$$

where $D_{Si}$ (cm$^2$ yr$^{-1}$) is the tortuosity- and bioturbation-corrected molecular diffusion coefficient of species $i$, $\alpha_i$ (yr$^{-1}$) is the bioirrigation coefficient, and $\Sigma R_i$ is the sum of biogeochemical reactions affecting $C_i$. Constitutive equations for $\phi$, $D_B$, $D_S$, $\omega_{acc}$ and $\alpha$ are given in Stolpovsky et al. (2015).

**Comment [SK13]:** To Rev. 2, Q. 11.

25   The unit conversion factor between POC and solutes due to mineralization reactions is:

$$n = \frac{\phi \cdot 12}{\rho \cdot (1-\phi(x))\cdot 10} \quad (18)$$

where $\rho$ (2.5 g cm$^{-3}$) is the dry sediment density.

Benthic fluxes of $O_2$ and $NO_3^-$ across the sediment surface were simulated and compared to a global database of $n$ = 185 observations. Porewater concentrations were further used for ground truthing the model, although far fewer data are

30   available.

**Comment [SK14]:** To Rev. 1, Q. 7.

Fixed concentrations were imposed for solutes (Dirichlet boundary) at the sediment surface ($x = 0$ cm). Measured bottom water concentrations were used for $O_2$, $NO_3^-$ and $NH_4^+$ whereas $NO_2^-$ and ODU were set to zero since they do not accumulate in the benthic boundary layer in ocean basins and margins. Flux continuity at the sediment surface for POC is equal to:

$$RRPOC_i = \omega_{acc}(wd) \cdot (1 - \phi(L)) \cdot POC_i(0,t) - \left(1 - \phi(0)\right) \cdot D_B(0) \left.\frac{\partial POC_i(x,t)}{\partial x}\right|_{x=0} \tag{19}$$

The upper boundary condition of each POC fraction was defined as a fraction of the rain rate:

$$RRPOC_i = f_i \cdot RRPOC \tag{20}$$

where $f_1$ and $f_0$ were derived from Eq. (11) and (12) and $f_2$ is equal to the CBE at each station. RRPOC was derived from the depth-integrated rate of POC degradation in the bioturbated layer, $RPOC_B$ (mmol m$^{-2}$ d$^{-1}$), and the CBE after solving the following equations:

$$\begin{cases} RRPOC - RRPOC \cdot CBE = RPOC_B \\ CBE = 0.013 + 0.53 \cdot \frac{RRPOC^2}{(7.0 + RRPOC)^2} \end{cases} \tag{21}$$

where CBE is calculated according to an empirical function that depends on rain rate (Dunne et al., 2007). $RPOC_B$ was approximated from a mass balance of the measured benthic fluxes of $O_2$, $NO_3^-$ and $NH_4^+$ (Stolpovsky et al., 2015).

At the bottom of the bioturbated layer ($x = 20$ cm), a zero gradient (Neumann) boundary was applied for all species. In reality, deeper anaerobic organic carbon mineralization will lead to non-zero concentration gradients for ODU and $NH_4^+$ at 20 cm. Yet, the overall contribution of anaerobic degradation below 20 cm is a small fraction of total degradation. Log-log relationships of radio-tracer measurements of sulfate reduction versus depth in continental marginal settings show that over 95 % of OM is mineralized in the upper 20 cm (Holmkvist et al., 2011). Imposing a zero-gradient condition for these species at the lower boundary is hence justifiable but introduces a small uncertainty in the model results. Further refinements to the model could be made by extending the approach to consider mineralization kinetics in deeper sediments. This is beyond the scope of this study, which focuses on POC mineralization in surface sediments only.

**Comment [SK15]:** To Rev. 1, Q. 8. And To Rev. 2, Q. 2.

**3 Results**

The rain rate of POC to the seafloor for each site in the database is compared with the Martin curve ($b = 0.86$) in Fig. 2a. The good agreement between the two demonstrates that the independent rain rate estimates based on benthic fluxes are largely consistent with the flux of sinking material based on open-ocean sediment trap observations. From the individual rain rates, CBE (Eq. 21), $k_{app}(wd)$ and the three rate constants for POC degradation, the proportion of reactive-to-highly reactive POC was determined for all ocean depths (Fig. 2b). With increasing water depth, the fraction of reactive POC, characterized by $k_1$, increases at the expense of the highly reactive fraction, characterized by $k_0$. This results from the more rapid mineralization of labile components in sinking organic detritus. $f_1/f_0$ increases from ~0 on the shelf to around 250 in the deep sea, meaning that < 1 % of reactive detritus is highly reactive by the time it reaches the deep ocean. This is to be expected because highly

reactive particles sinking with a speed of 50 m d$^{-1}$ require many weeks to reach deep ocean sediments, which compares to its turnover time of POC$_0$ of only ~1 week ($k_0$ = 70 yr$^{-1}$).

Simulated O$_2$ fluxes for the global database fit very well to measured data using the imposed $k_0$ and $k_1$ of 70 yr$^{-1}$ and 0.5 yr$^{-1}$, respectively (red symbols, Fig. 3a). Since O$_2$ is the ultimate electron acceptor, either for the direct oxidation of POC or for oxidation of the reducing metabolites of anaerobic respiration (i.e. ODUs), this goodness-of-fit demonstrates that the total carbon degradation rate in addition to the CBE are correctly simulated at each station. The NO$_3^-$ result shows more scatter (Fig. 3b). The benthic N cycle is convoluted with multiple sources and sinks extending over a large range of redox potential (Table 1 in Stolpovsky et al., 2015). Furthermore, the reaction rate constants of the N cycle are unknown at the global scale. At some stations, the model strongly underestimates the observed NO$_3^-$ flux. For example, the green ellipse corresponds to the Peruvian margin where NO$_3^-$ uptake is dominated by nitrate-storing sulfur oxidizing bacteria such as *Thioploca* spp. (Fossing et al., 1995; Dale et al., 2016). The current model does not consider this process.

Both O$_2$ and NO$_3^-$ fluxes are not particularly sensitive to 10-20 % variations in $k_0$ and $k_1$ (colored symbols on the plots). Although the NO$_3^-$ fluxes are more sensitive than O$_2$, no combination of these parameters is able to improve a fit at all the sites simultaneously, which can be put down to structural deficiencies in the model. Even so, the measured fluxes are reproducible with the model to within ca. 20 % over the prescribed variations in the rate constants. We take this to mean that the $k_{app}(wd)$ provides a robust constraint on the mineralization constants and the benthic fluxes, given the limits set by Eq. (14).

As a further validation of the approach, modeled and measured vertical geochemical profiles were compared for the same dataset as used to verify the power law model (Stolpovsky et al., 2015). These sites include diverse settings, such as the continental shelf (41 and 114 m water depth), upper slope (241 and 1025 m) and the deep-sea (3073 m) (Fig. 4). Two of the sites are located in high-nitrate-low-oxygen (HNLO) areas where poorly oxygenated waters bathe the seafloor (Washington and Mauritanian margin). The 3-G model (red curves) captures the observed trends in O$_2$ profiles through the bioturbated layer at these sites and also reproduces the trends in NO$_3^-$ concentrations. The prominent subsurface peaks in modeled O$_2$ and NO$_3^-$ at St. NH14A and St. H are apparently absent in the field data and probably caused by too intense bioirrigation in the model. The rate of this process is described using an empirical function based on total oxygen uptake and shows considerable uncertainty when applied globally (Meile and Van Cappellen, 2003). Nevertheless, these peaks do not affect O$_2$ and NO$_3^-$ gradients at the sediment surface, and the benthic fluxes are simulated accurately. Overall, the fluxes from the 3-G model are close to those from the power law model, which provides a continuous and arguably more realistic description of POC degradation. A closer correspondence with the 3-G versus 2-G model with power law is expected, because the multi-G model becomes more 'continuous' as the number of POC fractions increases. We would thus expect a model with even more fractions to perform better than the 3-G model, but with the added cost of more parameter uncertainty.

As a test of our assumption that three reactive size classes are required for simulating the data, we can compare our result with a 2-G model that includes only one reactive fraction. In this case, the rate constant for the single reactive fraction, $k^*$, can be obtained from the following equation:

**Comment [SK16]:** To Rev. 2, Q. 12.

**Comment [SK17]:** To Rev. 3, Q. 6.

$$(1 - \mathrm{CBE}) \cdot k^* + \mathrm{CBE} \cdot k_2 = k_{app}(wd) \tag{22}$$

where $k_2$ is defined in the same way as in the 3-G model and values of $k^*$ are intermediate between $k_0$ and $k_1$. The benthic fluxes derived with the 2-G model compare well with those derived with the 3-G model, although higher-end $NO_3^-$ fluxes tend to be overestimated relative to the 3-G model (Fig. 5). However, the porewater profiles using the 2-G model are very poorly simulated, especially for $NO_3^-$, due to the lack of carbon mineralization below the oxic layer where denitrification and anammox take place. The 2-G model is obviously structurally inferior to the 3-G model for simulating N sources and sinks accurately.

**Comment [SK18]:** To Rev. 3, Q. 5.

An alternative way to visualize this result is to compare the simulated $NO_3^-$ penetration depth (NPD) for an additional set of 12 stations that was used to validate the 1-D diagenetic model 'Muds' (Archer et al., 2002). NPD is defined as the sediment depth where $NO_3^-$ concentration falls to 2% of the local bottom water level. The NPD is a useful comparative metric because the depth-profile of organic matter degradation largely determines the depth where $NO_3^-$ is depleted. The 3-G model is able to predict the NPD at 10 from 12 stations (exceptions being #16 and #32) to within a factor of two or 1.5 cm, whereas for the 2-G model this decreases to 5 stations (Fig. 6). As before, the 3-G model is also closer to the power model result. This again demonstrates that one effective reactive POC pool defined by a single rate constant is insufficient to represent the reactivity of natural organic matter. At a minimum, two reactive fractions undergoing degradation on the time-scale of burial through the bioturbated layer, plus a poorly reactive fraction that is largely buried, are required to simulate the vertical structure of $O_2$ and $NO_3^-$ diagenesis in addition to POC burial.

**4 Discussion**

**4.1 Sedimentary organic carbon degradation constants**

As recently pointed out by Keil (2017), understanding the mechanisms and processes that determine the flux of organic carbon in the ocean interior (e.g., the biological pump) and preservation within sediments are among today's great oceanographic challenges (Doney and Karnauskas 2014; Heinze et al. 2015). Inclusion of organic carbon burial improves model distributions of redox sensitive tracers in the ocean such as oxygen and nutrients (Palastanga et al., 2011; Kriest and Oschlies, 2013) and, consequently, predictions of future climate scenarios. Proper parameterization of organic carbon degradation kinetics in sediments is thus of utmost importance for the forthcoming generation of ESMs that explicitly recycle and/or preserve sinking planktonic detritus in the benthic compartment.

In this study, we derived a 3-G diagenetic model that is able to simulate a global database of benthic $O_2$ and $NO_3^-$ fluxes and that is further consistent with global POC rain and burial rates (based on CBE) and porewater $O_2$ and $NO_3^-$ distributions. The opportunity to further ground-truth the model using fluxes of other solutes is limited by the severe lack of in situ measurements. The model does not perfectly simulate field data from site-specific locations (Figs. 3 and 4), which is probably caused by the empirical functions describing bioirrigation, bioturbation and CBE that themselves carry considerable uncertainties. Moreover, the model performs less well on the continental shelf due to the high sediment

heterogeneity there and because it is mainly tuned to the N cycle and lumps together coupled Fe-Mn-S cycling into ODUs. POC degradation coupled to the reduction of metal oxides and sulfate can be important in bioturbated sediments on the continental shelf (Canfield et al., 1993).

A novel aspect of the model is that the reactivity of the organic material degraded in the seafloor is linked to the apparent
5  reactivity of material sinking through the water column. It performs equally well as our previous approach that used a continuous power law to describe POC mineralization kinetics at the global scale (Stolpovsky et al., 2015). These workers showed that the apparent rate constant for POC mineralization can decrease by orders of magnitude through the bioturbated layer. This dynamic is reproduced with the 3-G model due to the dependence of the ratio of highly-reactive versus reactive organic matter on rain rate (Fig. 2). Whereas the power model is less suitable for settings with strong temporal variability in
10  POC rain rate (see Introduction), the current multi-G model is applicable in steady and non-steady state situations.

The motivation for this work is to provide a methodology for parametrizing POC degradation in ESMs of the global carbon cycle. In our approach, the corresponding rate constants of the highly reactive ($k_0 = 70$ yr$^{-1}$), reactive ($k_1 = 0.5$ yr$^{-1}$) and refractory ($k_2 = 0.001$ yr$^{-1}$) classes are fixed. However, the relative proportion of these classes varies globally as a function of the apparent reactivity of sinking POC, $k_{app}(wd)$. To apply this in a global model, one needs to know $k_{app}(wd)$ (Eq. 8). The
15  $k_{app}(wd)$ relationship is dependent on our choice of the $b$ coefficient in the Martin curve. Whilst not without its detractors (see below), a coefficient of 0.86 in the Martin curve seems to provide a robust constraint on reactivity of organic matter being degraded in the sediment (Fig. 2a). The model can be applied in ESMs that do not use the Martin curve as long as the $k_{app}(wd)$ of POC sinking to the seafloor can be recalculated.

The rate constants for $k_0$ and $k_1$ derived here correspond to POC that is degraded over days – years. This POC fraction is thus
20  mostly responsible for driving the coupling of sediment fluxes to water column processes in global biogeochemical models. Our proposed value of the highly reactive fraction $k_0$ (70 yr$^{-1}$), equivalent to a lifetime (1/$k$) of around 1 week, is toward the high end of those reported previously for $k_0$ using 'stand-alone' diagenetic models. In a review on this topic, Arndt et al. (2013) reported that values of $k_0$ in 3-G diagenetic models tuned to field data show a maximum of $10^1$ yr$^{-1}$. The highest value reported was 76 yr$^{-1}$, derived for an Arctic fjord using a carefully constrained empirical model (Berg et al., 2003). This
25  highly reactive pool was detectable from millimeter scale dissolved $O_2$ gradients obtained using microelectrodes. POC reactivity in most published benthic models is not constrained by finely resolved microelectrode data or in situ fluxes that constitute a robust constraint on the mineralization of the highly reactive POC fractions (Glud et al., 2009; Bohlen et al., 2011; Dale et al., 2016). The vertical resolution of porewater sampling using traditional extraction techniques will not capture sharp gradients in solute concentrations at the sediment–water interface resulting from the degradation of highly
30  reactive POC. Lower rate constants are typically derived by modeling the degradation of POC over larger sediment depths, manifested in the accumulation of ammonium or alkalinity for example (Berner, 1980).

Our rate constants are in agreement with the result of a well-documented two year-long decomposition experiment of fresh phytoplankton by Westrich and Berner (1984). These workers extracted a 3-G mineralization model from their data with rate constants of $24 \pm 4$, $1.4 \pm 0.7$ and 0 yr$^{-1}$ (Fig. 7). Our values are able to simulate the temporal decrease in phytoplankton

carbon with the same degree of accuracy despite the large differences in POC attenuation in the initial stages of the experiment ($R^2 = 0.98$ and standard error = 0.06 g/l in both cases). Whilst other incubation studies cited by Westrich and Berner (1984) provide similar rate constants, we were unable to simulate the $NO_3^-$ porewater profiles in our database using the given values. This may be because the sampling interval in experimental studies does not capture the degradation of POC

5   with very short lifetimes.

It is further noteworthy that our $k_0$ value, and those listed in Arndt et al. (2013), tend to be orders of magnitude higher than those used in ESMs that explicitly couple sediment and pelagic carbon cycles. For instance, rate constants for aerobic respiration in the 1-D benthic model coupled to HAMOCC range from 0.005 to 0.01 $yr^{-1}$ (Palastanga et al., 2011), whilst benthic organic matter mineralization in the MEDUSA model is associated with a turnover of 0.024 $yr^{-1}$ (Munhoven, 2007).

10  Sedimentary organic matter mineralization constants in ESMs are typically tuned to global organic carbon burial rates or sediment POC content (see Arndt et al., 2013). In general, POC content in surface sediments is a poor parameter to validate POC turnover because almost the entire organic matter raining to the deep-sea floor is degraded.

**4.2 Upscaling and uncertainties**

15  The multi-G model for benthic POC mineralization proposed here compartmentalizes organic matter into well packaged macromolecular fractions that obey first-order kinetics with rate constants that are fixed and independent of processes taking place in the sediment. This is, of course, a gross oversimplification of reality. Sedimentary organic matter is a complex array of chemical components characterized by large divergences in degradability and further altered by mineral interactions, geopolymerization, thermodynamic factors and the activity of micro-and macro-biota (e.g. de Leeuw and Largeau, 1993;

20  Aller, 1994; Mayer, 1995; LaRowe and Van Cappellen, 2011). These, plus other unknown and poorly understood factors (Burdige, 2007), lead to a continuous decrease in POC degradation over time, that is, with an apparent time-dependent rate constant (Middelburg et al., 1989). We are still far from a mechanistic understanding of the fundamental controls on organic matter degradation dynamics in diverse marine settings. Explicit formulations to capture the structural complexity of natural organic matter are currently beyond the practical capacity of diagenetic models, and the multi-G approach remains the

25  cornerstone of most diagenetic models published to date (Arndt et al., 2013). Clearly, then, the major controls on organic matter degradation and preservation can be encapsulated within the multi-G concept with considerable success. This is despite the fact that direct observations of POC turnover would tend to suggest that organic matter degradation rather follows kinetics best described with a power law model (Jørgensen and Parkes, 2010; Flury et al., 2016).

This observation implies that conceptually simple approaches like the multi-G model can be coupled to ESMs to account for

30  sediment feedbacks in simulations of climate-relevant processes. The need for benthic-pelagic coupling in ESMs has driven the search for a generalized and transferable framework for predicting sedimentary POC degradation kinetics at the regional and global scale (Stolpovsky et al., 2015). Yet, a review of the literature (>250 data sets) by Arndt et al. (2013), failed to

unearth any statistically robust relationship between the rate constants for organic matter degradation (both multi-G and continuum models) and major controlling factors including RRPOC, sedimentation rate and water depth. In fact, it would appear that the first-order rate constants are completely independent of these drivers (Arndt et al., 2013). This is consistent with our findings, where the $k$ values are fixed. Instead, we propose that an important consideration is the changing

5     *proportion* of the reactive classes reaching the seafloor in addition to their absolute reactivity. The partitioning of sinking material into discrete reactive classes is linked to the apparent reactivity of the sinking material itself, which we approximated using the Martin curve. Thus, the degradation of organic carbon in our model is consistent with, and set by, carbon degradation in the water column. With this top-down constraint, global trends in sedimentary benthic fluxes and porewater chemistry can be reproduced. Stolpovsky et al. (2015) arrived at a similar conclusion, where the reactivity of

10     organic matter described by a power law model was directly linked to the rain rate of organic material sinking out of the water column.

Inclusion of highly reactive POC fractions in diagenetic models imposes limits on the structural complexity of vertically-resolved benthic models coupled to global biogeochemical models. Diagenetic models are typically solved over hundreds of vertical sediment layers whose thickness increases from e.g. sub-mm scale at the sediment surface to e.g. decimeters or

15     meters in deeper strata. By comparison, and mainly due to computational constraints, benthic models coupled to ESMs tend to have a relatively low number of layers (e.g. 11 layers over the upper 10 cm in HAMOCC; Palastanga et al., 2011). Consider a 1-D advection-diffusion-reaction problem for the first-order decay of POC ($k = 70$ yr$^{-1}$) in a bioturbated layer (L = 10 cm) on the lower continental slope (2000 m). For these settings, $D_B = 0.5$ cm$^2$ yr$^{-1}$ and $\omega_{acc} = 0.005$ cm yr$^{-1}$ are typical (see references in Stolpovsky et al., 2015). The Peclet number is < 1 for these conditions, meaning that diffusive transport (bioturbation) dominates advective transport (burial). Further, for highly reactive POC, the characteristic time scale for POC

20     degradation is much smaller than for POC transport. To maintain model accuracy, therefore, the spatial discretization in the diagenetic model ($\Delta x$) must then be smaller than the relevant length scale, $(D_B/k)^{0.5}$, which in this example is less than 1 mm. This limitation is maintained throughout the ocean for typical values of $D_B$ and $\omega_{acc}$, requiring a stable routine for solving the diagenetic model (Berg et al., 2007). With a lower $k$ value of 0.01 yr$^{-1}$ as in the ESM mentioned above, the maximum $\Delta x$

25     increases significantly. These calculations are approximate since the dynamics of redox sensitive elements is typically non-linear, and here we only intend to demonstrate the need for higher spatial resolution to capture the fine-scaled distributions of sedimentary variables as $k$ increases. This is an important consideration for many processes, particularly for carbonate dissolution by respiratory $CO_2$ which takes place in the upper millimetres of the sediment (Hales, 2003; Jahnke and Jahnke, 2004). In general, for highly reactive classes, large reductions in the spatial resolution of the diagenetic model will lead to

30     inaccurate benthic feedbacks for redox sensitive elements such as $O_2$, $NO_3^-$, and $PO_4^{3-}$. Given these caveats and the computational penalties demanded by online coupling of diagenetic models to ESMs, it is understandable that global or regional models favour computationally efficient empirical transfer functions to simulate benthic feedbacks of N, P, Fe and $O_2$ cycles (Middelburg et al., 1997; Soetaert et al., 2000; Capet et al., 2016; Bohlen et al., 2012; Somes et al., 2013; Dale et al., 2015; Wallmann, 2010; Yang and Gruber, 2016).

**Comment [SK21]:** To Rev. 2, Q. 13.

The application of the multi-G model relies on knowledge of $k_{app}(wd)$ (Eq. 7) and the CBE (Eq. 21). Both of these are imperfectly known at the global scale. With regards to $k_{app}(wd)$, the depth attenuation of carbon flux in the ocean is critical to estimate the apparent reactivity of organic carbon raining to the seafloor. The Martin curve has found wide application in global models (e.g. Najjar et al., 1992; Ito and Follows, 2005). However, concerns about the accuracy of the Martin curve in capturing small-scale spatial and temporal variability of the biological pump in the upper ocean have called into question its suitability for global scale applications (Buessler et al., 2007). A rather wide range of the $b$ coefficient between 0.4 and 1.75 has been reported and discussed at length (Keil, 2017; Guidi et al., 2015 and references therein). The sinking POC flux also varies temporally as well as spatially in the open ocean (Henson et al., 2015). Certain components of biological pump, such a mineral ballasting or diatom coagulation, can rapidly deliver fresh organic matter to the ocean interior (Buesseler et al., 2008; Belcher et al., 2016). To a first approximation, however, the Martin curve with a $b = 0.86$ captures much of the independent rain rates estimates in our database (Fig. 2a). These rain rates were derived from benthic fluxes that integrate POC degradation over much longer periods of time and so provide a more average impression of how carbon rain varies with water depth. This supports the use of $k_{app}(wd)$ derived from the classical Martin curve in benthic studies. A ±20 % change in $b$ leads to only minor changes in the benthic flux. For instance, for St. NH14A (114 m) in Fig. 4, the flux of $O_2$ calculated with $b = 0.69$ differs from the one calculated with $b = 1.03$ by 14% and $NO_3^-$ by 13.5%. For St. PS236-1 (3073 m), the change is only 0.02% and 0.13% respectively.

**Comment [SK22]:** To Rev. 3, Q. 1., part 1.

Trends in carbon burial efficiencies are generally understood throughout the ocean: low CBEs characterize the deep-sea (< 5 %) versus higher CBEs on the margins (>20 %). Yet, data are limited and subject to error due to mismatches in the time-scales of input parameters needed to determine CBE (Burdige, 2007). Additionally, the mechanistic controls on CBE are still not resolved (Hedges and Keil, 1995). Although the Dunne relationship given in Eq. (21) does not perfectly describe rain rates at the global scale, it is consistent with previous studies and empirical models based on global POC burial rates (e.g. Flögel et al., 2011; Kriest and Oschlies, 2013). The good correspondence between measured and observed $O_2$ fluxes (Fig. 3a) indicates that the CBE formulation by Dunne et al. (2007) is appropriate to use in global models.

**Comment [SK23]:** To Rev. 3, Q. 1., part 2.

Finally, an additional source of uncertainty rests with the sinking speed of particulate organic matter through the water column and the distribution of particle sizes, both of which were used to derive the apparent reactivity of settling POC (see Methods). These were parameterized according to the NEMO-PISCES model that is tuned to biogeochemical tracers in the water column as well as global benthic $O_2$ fluxes (Aumont et al., 2017). Whilst it is beyond the scope of the present study to discuss the uncertainty in these model aspects, it worth noting that the contribution of large POC particles to total POC in NEMO-PISCES (assumed here to be 20 %) does vary, particularly on the margins. For instance, larger particles contribute up to 50 % of the total in eastern boundary systems and areas of higher primary production (O. Aumont, pers. comm). Since larger particles will tend to sink more quickly, the flux of POC to the seafloor in these regions will be greater than predicted by the Martin curve (i.e. $b < 0.86$). However, parameterizing the sinking velocities is evidently not straightforward since they span several orders of magnitude (meters to thousands of meters per day) (Kriest and Oschlies, 2008). Even small changes in the sinking speed or remineralization rates can change the $e$-folding depth for carbon export by hundreds of meters (DeVries

et al., 2012). Ecosystem structure, and the associated packaging, disaggregation, and repackaging processes during settling may create more spatial and temporal variability in POC sinking flux, hence its apparent reactivity, than assumed by a single *b* coefficient (De La Rocha and Passow, 2007; Wilson et al. 2008; Henson et al., 2012). Moreover, lateral transport of POC on the continental margins to deep-sea depocenters can be accelerated in benthic and intermediate nepheloid layers and by

5   mass wasting events (Jahnke et al., 1990; Thomsen, 1999; Inthorn et al., 2006). A quantitative accounting of the contribution of lateral POC transport to the biological pump and preservation in sediments must surely be one of the major outstanding challenges in ocean science.

**5 Conclusion**

10   Biogenic elements (C, N, P, Fe) that are packaged into organic material in the sunlit ocean are either mineralized in the ocean interior or deposited on the seafloor. Most organic material reaching the seafloor is mineralized and returned to the ocean. The rest is permanently lost from the bioavailable pool either by microbial activity (e.g. N loss to $N_2$ by denitrification) or through mineral precipitation and/or burial. Over long time-scales, these sediment sinks begin to exert a major control on nutrient inventories, ocean fertility, and climate-ocean interactions. Realistic model simulations of ocean

15   chemistry in past, present and future scenarios must account for the recycling of organic material at the seafloor.

In an attempt to improve our understanding of the rate and depth of organic carbon mineralization in seafloor surface sediments, we parameterized a 1-D diagenetic model for organic carbon mineralization that can be coupled to Earth System Models. In contrast to previous approaches, the apparent reactivity of the organic material degraded in the seafloor is continuous with, and set by, the apparent reactivity of material sinking through the water column. We propose that an

20   important and mostly overlooked consideration in previous upscaling approaches is the *proportion* of the reactive fractions classes reaching the seafloor in addition to their intrinsic reactivity. The results imply the presence of a highly reactive organic carbon class at the sediment water interface with a turnover time of days to weeks. Mineralization of this pool will have important implications for the flux of redox sensitive elements to/from the ocean as well as for carbonate dissolution by respiratory $CO_2$.

[revised manuscript text omitted]